# When Do Prompting and Prefix-Tuning Work? A Theory of Capabilities and Limitations

**Aleksandar Petrov, Philip H.S. Torr & Adel Bibi**
Department of Engineering Science
University of Oxford
Oxford, United Kingdom
`{aleksandar.petrov,philip.torr,adel.bibi}@eng.ox.ac.uk`

## Abstract

Context-based fine-tuning methods, including prompting, in-context learning, soft prompting (also known as prompt tuning), and prefix-tuning, have gained popularity due to their ability to often match the performance of full fine-tuning with a fraction of the parameters. Despite their empirical successes, there is little theoretical understanding of how these techniques influence the internal computation of the model and their expressiveness limitations. We show that despite the continuous embedding space being more expressive than the discrete token space, soft-prompting and prefix-tuning are potentially less expressive than full fine-tuning, even with the same number of learnable parameters. Concretely, context-based fine-tuning cannot change the relative attention pattern over the content and can only bias the outputs of an attention layer in a fixed direction. This suggests that while techniques like prompting, in-context learning, soft prompting, and prefix-tuning can effectively elicit skills present in the pretrained model, they may not be able to learn novel tasks that require new attention patterns.

## 1 Introduction

Language model advances are largely driven by larger models and more training data (Kaplan et al., 2020; Rae et al., 2021). Training cutting-edge models is out of reach for most academic researchers, small enterprises, and individuals, and it has become common to instead fine-tune open-source pretrained models (Devlin et al., 2019; Min et al., 2021). Yet, due to escalating computational demands, even fine-tuning of the larger models has become prohibitively expensive (Lialin et al., 2023).

As a result, there is an acute need for more efficient fine-tuning methods, either by sparsely modifying the parameters of the model or modifying its input context. Examples of the first type include adapter modules which introduce a few trainable layers to modify the behaviour of the frozen pretrained network (Rebuffi et al., 2017; Houlsby et al., 2019; Hu et al., 2023). One can also use low-rank updates, which also results in a reduced number of trainable parameters (Hu et al., 2021).

Context-based fine-tuning has been motivated by the success of few-shot and zero-shot learning (Wei et al., 2021; Kojima et al., 2022). The most popular context-based approach is prompting, where generation is conditioned on either human-crafted or automatically optimized tokens (Shin et al., 2020; Liu et al., 2023). In-context learning —prompting via providing input-label examples— is another widely used technique (Brown et al., 2020). Given the challenges of discrete optimization over tokens, there is a growing interest in methods that optimize real-valued embeddings (Lester et al., 2021). It is widely believed that these *soft prompts* offer greater expressiveness due to the expansive nature of continuous space. Furthermore, beyond only optimizing input embeddings, one can optimize the inputs of every attention layer (Li and Liang, 2021). This technique, *prefix-tuning*, has proven to be very successful and competitive to full fine-tuning (Liu et al., 2022).

While context-based fine-tuning approaches have witnessed impressive empirical successes and widespread adoption, we have little theoretical understanding of how they work. In this work, we analyse the influence of prompts and prefixes on the internal computations of a pretrained model and delineate their limitations. Specifically, we address the following questions:

1. **Soft prompting and prefix-tuning are motivated by the embedding space being larger than the token space. However, can a transformer utilize the additional capacity?** We show that with a careful choice of transformer weights, controlling a single embedding can generate any of the $V^N$ completions of $N$ tokens, while controlling a token can produce only $V$ completions, with $V$ being the vocabulary size. Thus, a transformer can indeed exploit the embedding space.

2. **Since prefix-tuning is more expressive than prompting, is it as expressive as full fine-tuning?** Despite the expressiveness of continuous space, prefix-tuning has structural limitations. A prefix cannot change the relative attention over the content tokens and can only bias the output of the attention block in a constant direction. In contrast, full fine-tuning can learn new attention patterns and arbitrarily modify attention block outputs, making it strictly more powerful.

3. **If context-based fine-tuning methods suffer from such structural limitations, how come they have high empirical performance?** We show that the prefix-induced bias can steer the model towards a pretraining task. Prefix-tuning can also combine skills picked up during pretraining to solve some new tasks similar to pretraining tasks. However, it may not learn a completely new task. This is not simply because of the small number of learnable parameters: fine-tuning the same number of parameters can be sufficient to learn the novel task. Hence, context-based fine-tuning can elicit or combine pretrained model skills but cannot learn completely new behaviors.

## 2 BACKGROUND

### 2.1 THE TRANSFORMER ARCHITECTURE

We outline a simplified decoder-only transformer architecture (Vaswani et al., 2017). Assume that the model has vocabulary size $V$ (also referred to as *number of tokens*). The input is a sequence $(\mathrm{x}_1, \ldots, \mathrm{x}_p)$, $\mathrm{x}_i \in \{1, \ldots, V\}$, $\forall i$. Each token is mapped to a $d_e$-dimensional vector that is the $\mathrm{x}_i$-th column of an embedding matrix $\boldsymbol{E} \in \mathbb{R}^{d_e \times V}$. The attention mechanism is position-invariant, so typically position encodings are added. For a model with maximum input length $N$ (*context size*), we use a one-hot position encoding $\boldsymbol{e}_N(i)$ concatenated with the embedding. Therefore, the embedding for the $i$-th position provided to the first attention block would be $\boldsymbol{x}_i = [\boldsymbol{E}_{:,\mathrm{x}_i}^\top, \boldsymbol{e}_N^\top(i)]^\top$.

A transformer consists of alternating attention blocks which operate across the whole sequence and Multi-Layer Perceptrons (MLPs) that operate on each individual element. Each attention block consists of $H$ heads. Each head $h$ is parameterized by query, key, and value matrices $\boldsymbol{W}_Q^h, \boldsymbol{W}_K^h \in \mathbb{R}^{k \times d_{\text{in}}}, \boldsymbol{W}_V^h \in \mathbb{R}^{d_{\text{out}} \times d_{\text{in}}}$.[1] The attention matrix $\boldsymbol{A}^h \in \mathbb{R}^{p \times p}$ for head $h$ then has elements

$$\boldsymbol{A}_{ij}^h = \frac{\exp\left(T/\sqrt{k}(\boldsymbol{W}_Q^h \boldsymbol{x}_i)^\top (\boldsymbol{W}_K^h \boldsymbol{x}_j)\right)}{\sum_{r=1}^p \exp\left(T/\sqrt{k}(\boldsymbol{W}_Q^h \boldsymbol{x}_i)^\top (\boldsymbol{W}_K^h \boldsymbol{x}_r)\right)}, \tag{1}$$

where $p \leq N$ is the current length of the input and $T > 0$ is an inverse temperature parameter.[2] Equation (1) is the softmax function, hence with high enough $T$, it will result in approximately one-hot encoding of the maximum $j$. The output of the attention block $\mathcal{A}$ with $H$ heads is then $(\boldsymbol{t}_1, \ldots, \boldsymbol{t}_p)$, where each position $i$ is the sum of the attention-weighted values across all heads:

$$\mathcal{A}[(\boldsymbol{W}_Q^1, \ldots, \boldsymbol{W}_Q^H), (\boldsymbol{W}_K^1, \ldots, \boldsymbol{W}_K^H), (\boldsymbol{W}_V^1, \ldots, \boldsymbol{W}_V^H)](\boldsymbol{x}_1, \ldots, \boldsymbol{x}_p) = (\boldsymbol{t}_1, \ldots, \boldsymbol{t}_p),$$
$$\boldsymbol{t}_i = \sum_{h=1}^H \sum_{j=1}^p \boldsymbol{A}_{ij}^h \boldsymbol{W}_V^h \boldsymbol{x}_j. \tag{2}$$

A transformer then applies an MLP to each output of an attention block before passing them to next attention block. We will consider linear layers $\mathcal{L}[\boldsymbol{M}, \boldsymbol{b}](\boldsymbol{x}) = \boldsymbol{M}\boldsymbol{x} + \boldsymbol{b}$ and ReLU-activated linear layers $\hat{\mathcal{L}}[\boldsymbol{M}, \boldsymbol{b}](\boldsymbol{x}) = \text{ReLU}(\boldsymbol{M}\boldsymbol{x} + \boldsymbol{b})$. When we compose attention blocks and linear or softmax layers, we will implicitly assume that the linear layer is applied to all positions of the sequence. Furthermore, we will use the *then* operator $\mathbin{\text{⨾}}$ for left-to-right function composition. Therefore, a transformer model predicting confidences over the vocabulary can, for example, be represented as:

$$(\boldsymbol{y}_1, \ldots, \boldsymbol{y}_p) = \left(\mathcal{A}_1 \mathbin{\text{⨾}} \hat{\mathcal{L}}_{1,1} \mathbin{\text{⨾}} \mathcal{L}_{1,2} \mathbin{\text{⨾}} \mathcal{A}_2 \mathbin{\text{⨾}} \hat{\mathcal{L}}_{2,1} \mathbin{\text{⨾}} \mathcal{L}_{2,2} \mathbin{\text{⨾}} \text{softmax}\right) \left(\begin{bmatrix} \boldsymbol{E}_{:,\mathrm{x}_1} \\ \boldsymbol{e}_N(1) \end{bmatrix}, \ldots, \begin{bmatrix} \boldsymbol{E}_{:,\mathrm{x}_p} \\ \boldsymbol{e}_N(p) \end{bmatrix}\right), \tag{3}$$

---

[1] For the first block, $d_{\text{in}}$ must be $d_e + N$ but may be different for the deeper blocks.

[2] A causal model has $A_{ij} = 0$ for $j > i$. This does not affect our results so we will skip the masking step.

where the output dimension of the last layer has to be $V$. The next token for a deterministic transformer is selected to be the last element's largest logit: $\mathbf{x}_{p+1} = \arg\max_{u \in 1,...,V} \boldsymbol{y}_{p,u}$. Given an input $(\mathbf{x}_1, \ldots, \mathbf{x}_p)$, the model then autoregressively extends this sequence one token at a time, following Equation (3) either until the sequence reaches a length $N$ or until a special termination token.

A transformer has no separation between the system prompt S, user provided input X and the autoregressively response Y. Thus, a sequence conditional on user input is denoted as $(\mathbf{S}_1, ..., \mathbf{S}_{n_S}, \mathbf{X}_1, ..., \mathbf{X}_{n_X}, \mathbf{Y}_1, ..., \mathbf{Y}_{n_Y})$ and one without user input as $(\mathbf{S}_1, ..., \mathbf{S}_{n_S}, \mathbf{Y}_1, ..., \mathbf{Y}_{n_Y})$.

## 2.2 Context-based fine-tuning of a pretrained model

We now define prompting, soft prompting and prefix-tuning with the previously introduced notation.

**Prompting.** The most frequently used content-based fine-tuning approach is *prompting*: prefixing the input $(\mathbf{X}_1, ..., \mathbf{X}_{n_X})$ with a token sequence $S \in \{1, ..., V\}^{n_S}$ to guide the model response: $(\mathbf{S}_1, ..., \mathbf{S}_{n_S}, \mathbf{X}_1, ..., \mathbf{X}_{n_X})$. This is how most people interact with language models such as ChatGPT.

**Soft prompting.** Soft prompting replaces the embeddings of the system input $\boldsymbol{E}_{:,\mathbf{S}_i}$ with learned vectors $\boldsymbol{s}_i \in \mathbb{R}^{d_e}$ called *virtual tokens* (Hambardzumyan et al., 2021; Lester et al., 2021; Qin and Eisner, 2021). Hence, the input in Equation (3) is modified to be:

$$\left( \begin{bmatrix} \boldsymbol{s}_1 \\ \boldsymbol{e}_N(1) \end{bmatrix}, \ldots, \begin{bmatrix} \boldsymbol{s}_{n_S} \\ \boldsymbol{e}_N(n_S) \end{bmatrix}, \begin{bmatrix} \boldsymbol{E}_{:,\mathbf{X}_1} \\ \boldsymbol{e}_N(n_S + 1) \end{bmatrix}, \ldots, \begin{bmatrix} \boldsymbol{E}_{:,\mathbf{X}_{n_X}} \\ \boldsymbol{e}_N(n_S + n_X) \end{bmatrix} \right) \tag{4}$$

with $\boldsymbol{s}_i$ chosen to maximize the likelihood of a target response $Y = (\mathbf{Y}_1, ..., \mathbf{Y}_{n_Y})$, i.e., $\arg\max_{\boldsymbol{s}_1,...,\boldsymbol{s}_{n_S} \in \mathbb{R}^{d_e}} \sum_{j=1}^{n_Y} \log \boldsymbol{y}_{n_S + n_X + j, \mathbf{Y}_j}$, where $\boldsymbol{y}_{n_S + n_X + j}$ are autoregressively generated.

**Prefix-tuning.** Prefix-tuning applies soft prompting across the depth of the model (Li and Liang, 2021; Liu et al., 2022). The first $n_S$ positions for all attention blocks are learnable parameters, replacing the input $(\boldsymbol{x}_1^l, \ldots, \boldsymbol{x}_{n_X}^l)$ for layer $l$ with $(\boldsymbol{s}_1^l, \ldots, \boldsymbol{s}_{n_S}^l, \boldsymbol{x}_1^l, \ldots, \boldsymbol{x}_{n_X}^l)$, where all $\boldsymbol{s}_i^l$ constitute the prefix. Hence, prefix-tuning can be formulated as $\arg\max_{\{\boldsymbol{s}_1^1,...,\boldsymbol{s}_i^L\}_{i=1}^{n_S}} \sum_{j=1}^{n_Y} \log \boldsymbol{y}_{n_S + n_X + j, \mathbf{Y}_j}$. Prefix-tuning has been successful at fine-tuning models (Vu et al., 2022; Wu and Shi, 2022; Choi and Lee, 2023; Ouyang et al., 2023; Bai et al., 2023), leading to calls for language models provided as a service (La Malfa et al., 2023) to allow providing prefixes instead of prompts (Sun et al., 2022).

Any token-based prompt $(\mathbf{S}_1, ..., \mathbf{S}_{n_S})$ has a corresponding soft prompt $(\boldsymbol{s}_i = \boldsymbol{E}_{:,\mathbf{S}_i})$ but the reverse does not hold. Similarly, every soft prompt $(\boldsymbol{s}_1, ..., \boldsymbol{s}_{n_S})$ can be represented as a prefix by setting the deeper prefixes to be the values that the model would compute at these positions ($\boldsymbol{s}_i^l = (\mathcal{A}_1 \; \r\!\!\; ... \; \r\!\!\; \mathcal{L}_{l-1,-1})([\boldsymbol{s}_1^\top, \boldsymbol{e}_N(1)^\top]^\top, ..., [\boldsymbol{s}_l^\top, \boldsymbol{e}_N(l)^\top]^\top)$). The reverse also does not hold: there are prefixes that cannot be represented as a soft prompt. A hierarchy emerges: *prompting < soft prompting < prefix-tuning*, with prefix-tuning the most powerful of the three. Hence, we focus on examining its performance relative to full fine-tuning but our findings also apply to prompting and soft prompting.

## 3 Soft prompting has more capacity than prompting

The success of soft prompting (and prefix-tuning) is commonly attributed to the larger capacity of the continuous embeddings compared to the finite tokens. Yet, increased capacity is beneficial only if the model can utilize it. We show this is indeed the case by constructing a transformer generating exponentially more completions by varying a single virtual token than by varying a hard token.

Consider unconditional generation (representing a function with no inputs) with a single system token: $(\mathbf{Y}_1, ..., \mathbf{Y}_N) = f(\mathbf{S}_1) = f_{\mathbf{S}_1}$. For a deterministic autoregressive function, there are a total of $V$ functions in this family, hence the upper bound on the number of outputs of length $N$ that one can generate by varying the first token $\mathbf{S}_1$ is $V$: the first token fully determines the rest of the sequence. Generally, if one varies the first $N_S$ tokens, there are at most $V^{N_S}$ unique outputs. What if instead of the token $\mathbf{S}_1$ we vary a single virtual token $\boldsymbol{s}_1$: $(\mathbf{Y}_1, ..., \mathbf{Y}_N) = f(\boldsymbol{s}_1) = f_{\boldsymbol{s}_1}$? This family of functions is indexed by a real vector and hence is infinite: in principle, one could generate all $V^N$ possible

output sequences by only controlling $s_1$.[3] Still, a transformer may not be able to represent a function that achieves that in practice, i.e., it is not obvious if there is a surjective map from $\{f_{s_1} : s_1 \in \mathbb{R}^{d_e}\}$ to $\{1, ..., V\}^N$. We show that, in fact, there is a transformer $f$ for which such a surjective map exists:

**Theorem 1** (Exponential unconditional generation capacity of a single virtual token). *For any $V, N > 0$, there exists a transformer with vocabulary size $V$, context size $N$, embedding size $d_e = N$, one attention layer with two heads and a three-layer MLP such that it generates any token sequence $(Y_1, ..., Y_N) \in \{1, ..., V\}^N$ when conditioned on the single virtual token $s_1 = \left((Y_1-1)/V, ..., (Y_N-1)/V\right)$.*

However, conditional generation is more interesting: given a user input $(X_1, ..., X_{n_X})$, we want to generate a target response $(Y_1, ..., Y_{n_Y})$. Even in the simple case of one system token, the user provides one token and the model generates one token in response $(Y_1 = f(S_1, X_1) = f_{S_1}(X_1))$, we cannot control response of the model to any user input with the system token. There are $V^V$ maps from $X_1$ to $Y_1$, but $S_1$ can take on only $V$ values: $|\{f_{S_1} : S_1 \in 1, ..., V\}| = V < V^V$. Hence, tokens cannot be used to specify an arbitrary map from user input to model output. However, a single virtual token can specify any of the $V^V$ maps, i.e., there exists a transformer $f_{s_1}(X_2)$ for which there is a surjective map from $\{f_{s_1} : s_1 \in \mathbb{R}^{d_e}\}$ to $\{1, ..., V\}^{\{1, ..., V\}}$.

**Theorem 2** (Conditional generation capacity for a single virtual token ($n_X = n_Y = 1$)). *For any $V > 0$, there exists a transformer with vocabulary size $V$, context size $N = 2$, embedding size $d_e = V$, one attention layer with two heads and a three-layer MLP that reproduces any map $m: [1, ..., V] \rightarrow [1, ..., V]$ from a user input token to a model response token when conditioned on a single virtual token $s_1 = (m(1)/V, ..., m(V)/V)$. That is, by selecting $s_1$ we control the model response to any user input.*

Theorem 2 builds on Theorem 1 by showing that soft prompting is also more expressive for governing the *conditional* behavior of a transformer model. This also holds for longer responses $n_Y > 1$ by increasing the length of the soft prompt, or longer user inputs $n_X > 1$, by increasing the depth of the model. We provide proofs in Appendix A, as well as working Python implementations.

This section showed that soft prompting, and by implication, prefix-tuning, possess greater expressiveness than prompting. As we can fully determine the map from user input to model response using virtual tokens, our findings may appear to suggest that soft prompting is as powerful as full fine-tuning. However, this is not at all the case. There are structural constraints on the capabilities of soft prompting and prefix-tuning; they cannot facilitate the learning of an entirely new task. The following section elucidates this discrepancy and reconciles these seemingly contradictory results.

## 4 PREFIX-TUNING CAN ONLY BIAS THE OUTPUT OF AN ATTENTION HEAD

We just saw that soft prompting and prefix-tuning can fully control the conditional behavior of a transformer. However, that assumed a specific design for the network weights. Given a fixed pretrained model, as opposed to a manually crafted one, can prefix-tuning be considered equally powerful to full fine-tuning? In this section, we show that, for an arbitrary pretrained model, a prefix $S$ cannot change the relative attention over the content $X, Y$ and can only bias the attention block outputs in a subspace of rank $n_S$, the prefix length, making it less powerful than full fine-tuning.

**While full fine-tuning can alter the attention pattern of an attention head, prefix-tuning cannot.** Recall the attention $A_{ij}$ position $i$ gives to position $j$ for a trained transformer (Equation (1)):

$$A_{ij} = \frac{\exp\left(T/\sqrt{k}\, \boldsymbol{x}_i^\top \boldsymbol{W}_Q^\top \boldsymbol{W}_K \boldsymbol{x}_j\right)}{\sum_{r=1}^p \exp\left(T/\sqrt{k}\, \boldsymbol{x}_i^\top \boldsymbol{W}_Q^\top \boldsymbol{W}_K \boldsymbol{x}_r\right)} = \frac{\exp\left(T/\sqrt{k}\, \boldsymbol{x}_i^\top \boldsymbol{H} \boldsymbol{x}_j\right)}{\sum_{r=1}^p \exp\left(T/\sqrt{k}\, \boldsymbol{x}_i^\top \boldsymbol{H} \boldsymbol{x}_r\right)}, \tag{5}$$

where $\boldsymbol{W}_Q^\top \boldsymbol{W}_K = \boldsymbol{H}$. Full fine-tuning can enact arbitrary changes to $\boldsymbol{W}_Q$ and $\boldsymbol{W}_K$ and hence, assuming the input does not change (e.g., at the first attention layer), we get the following attention:

$$A_{ij}^{\text{ft}} = \frac{\exp\left(T/\sqrt{k}\, \boldsymbol{x}_i^\top \boldsymbol{H} \boldsymbol{x}_j + T/\sqrt{k}\, \boldsymbol{x}_i^\top \Delta \boldsymbol{H} \boldsymbol{x}_j\right)}{\sum_{r=1}^p \exp\left(T/\sqrt{k}\, \boldsymbol{x}_i^\top \boldsymbol{H} \boldsymbol{x}_r + T/\sqrt{k}\, \boldsymbol{x}_i^\top \Delta \boldsymbol{H} \boldsymbol{x}_r\right)},$$

---

[3]For example, LLaMA-7B (Touvron et al., 2023) has 24 426 unique completions when prompted with each of its 32 000 tokens and we found a non-exhaustive set of 46 812 unique 10-token-long sequences by controlling the first virtual token. Hence, in practice, one can generate more outputs by soft prompting than by prompting.

where the changes to $\boldsymbol{W}_Q$ and $\boldsymbol{W}_K$ are folded into $\Delta\boldsymbol{H}$. It is clear that by varying $\Delta\boldsymbol{H}$ full fine-tuning can change the attention patterns arbitrarily. However, let us see how is attention affected by the presence of a prefix. For now, assume we have a prefix of length one ($\boldsymbol{s}_1$) at position 0.

$$\boldsymbol{A}_{i0}^{\text{pt}} = \frac{\exp\left(T/\sqrt{k}\ \boldsymbol{x}_i^\top\boldsymbol{H}\boldsymbol{s}_1\right)}{\exp\left(\frac{T}{\sqrt{k}}\ \boldsymbol{x}_i^\top\boldsymbol{H}\boldsymbol{s}_1\right)+\sum\limits_{r=1}^{p}\exp\left(\frac{T}{\sqrt{k}}\ \boldsymbol{x}_i^\top\boldsymbol{H}\boldsymbol{x}_r\right)}, \quad \boldsymbol{A}_{ij}^{\text{pt}} = \frac{\exp\left(T/\sqrt{k}\ \boldsymbol{x}_i^\top\boldsymbol{H}\boldsymbol{x}_j\right)}{\exp\left(\frac{T}{\sqrt{k}}\ \boldsymbol{x}_i^\top\boldsymbol{H}\boldsymbol{s}_1\right)+\sum\limits_{r=1}^{p}\exp\left(\frac{T}{\sqrt{k}}\ \boldsymbol{x}_i^\top\boldsymbol{H}\boldsymbol{x}_r\right)} \text{ for } j\geq 1.$$

The numerator of $\boldsymbol{A}_{ij}^{\text{pt}}$ is the same as in Equation (5), i.e., the prefix does not affect it. It only adds the term $\exp(T/\sqrt{k}\ \boldsymbol{x}_i^\top\boldsymbol{H}\boldsymbol{s}_1)$ to the denominator. Therefore, the attention position $i$ gives to the content positions $j\geq 1$ is simply scaled down by the attention it now gives to the prefix. If *tomato* attends the most to *salad* in a particular context, no prefix can change that. This becomes evident by rewriting $\boldsymbol{A}_{ij}^{\text{pt}}$ as the attention of the pretrained model scaled by the attention "stolen" by the prefix:

$$\boldsymbol{A}_{ij}^{\text{pt}} = \boldsymbol{A}_{ij}\sum\nolimits_{r=1}^{p}\boldsymbol{A}_{ir}^{\text{pt}} = \boldsymbol{A}_{ij}(1 - \boldsymbol{A}_{i0}^{\text{pt}}). \tag{6}$$

Hence, prefix-tuning cannot affect the relative attention patterns across the content, it will only scale them down. In other words, one cannot modify what an attention head attends to via prefix-tuning.[4]

**Prefix-tuning only adds a bias to the attention block output.**   Let us see how this attention scaling down affects the output of the attention block. Following Equation (2), the output at position $i$ for the pretrained ($\boldsymbol{t}_i$), the fully fine-tuned ($\boldsymbol{t}_i^{\text{ft}}$) and the prefix-tuned ($\boldsymbol{t}_i^{\text{pt}}$) models are as follows:[5]

$$\boldsymbol{t}_i = \sum\nolimits_{j=1}^{p}\boldsymbol{A}_{ij}\boldsymbol{W}_V\boldsymbol{x}_j, \qquad\qquad \boldsymbol{t}_i^{\text{ft}} = \sum\nolimits_{j=1}^{p}\boldsymbol{A}_{ij}^{\text{ft}}(\boldsymbol{W}_V + \Delta\boldsymbol{W}_V)\boldsymbol{x}_j,$$

$$\boldsymbol{t}_i^{\text{pt}} = \boldsymbol{A}_{i0}^{\text{pt}}\boldsymbol{W}_V\boldsymbol{s}_1 + \sum_{j=1}^{p}\boldsymbol{A}_{ij}^{\text{pt}}\boldsymbol{W}_V\boldsymbol{x}_j \overset{(6)}{=} \boldsymbol{A}_{i0}^{\text{pt}}\boldsymbol{W}_V\boldsymbol{s}_1 + \sum_{j=1}^{p}\boldsymbol{A}_{ij}(1-\boldsymbol{A}_{i0}^{\text{pt}})\boldsymbol{W}_V\boldsymbol{x}_j = \boldsymbol{A}_{i0}^{\text{pt}}\boldsymbol{W}_V\boldsymbol{s}_1 + (1-\boldsymbol{A}_{i0}^{\text{pt}})\boldsymbol{t}_i. \tag{7}$$

Hence, prefix-tuning only biases the attention block value at each position $i$ towards the constant vector $\boldsymbol{W}_V\boldsymbol{s}_1$, which is independent of the content ($\boldsymbol{x}_1,...,\boldsymbol{x}_p$). I.e., the prefix-tuned activation is a linear combination of the pretrained activation and the constant vector $\boldsymbol{W}_V\boldsymbol{s}_1$. The content only affects the scale $\boldsymbol{A}_{i0}^{\text{pt}}$ of the bias via the amount of attention on the prefix. In contrast, in full fine-tuning $\Delta\boldsymbol{W}_Q$, $\Delta\boldsymbol{W}_K$ and $\Delta\boldsymbol{W}_V$ allow for a content-dependent change of the attention and value computation. These results hold for suffix-tuning (placing the prefix after the input) but *not* for suffix soft-prompting. We validate that this indeed is the case when prefix-tuning real-world transformers. In Figures 5 and 6, we show that a prefix applied to LLaMA's first layer does not change the relative attention distribution over the content positions $X$ and results in a bias with a constant direction.

**Longer prefixes define larger subspaces for the bias but are not fully utilized in practice.**   In the case of a longer prefix ($\boldsymbol{s}_1,...,\boldsymbol{s}_{n_S}$), the bias vector is in a subspace of dimensionality $n_S$: $\boldsymbol{t}_i^{\text{pt}} = \sum_{j=1}^{n_S}\boldsymbol{A}_{i,S_j}^{\text{pt}}\boldsymbol{W}_V\boldsymbol{s}_j + (1 - \sum_{j=1}^{n_S}\boldsymbol{A}_{i,S_j}^{\text{pt}})\boldsymbol{t}_i$, where $i$ goes over the content and $j$ over the prefix positions. Larger prefixes thus have a larger subspace to modify the attention block output. The specific direction is determined by the relative distribution of attention across the prefix positions. However, when we examine the distribution of attention across the prefix positions for various inputs as in Appendix B, it appears that the prefixes do not span this subspace. Regardless of the input, the attention $\boldsymbol{A}_{i,S_j}^{\text{pt}}$ over the prefix positions remains nearly constant. Thus, prefix-tuning does not seem to make full use of the space that the vectors $\boldsymbol{W}_V\boldsymbol{s}_j$ span. We hypothesise that this is due to the two competing optimization goals for the vectors $\boldsymbol{s}_j$: at the same time they need to "grab attention" when interacting with $\boldsymbol{W}_K$ and determine the bias direction when multiplied with $\boldsymbol{W}_V$.

**So, is prefix-tuning equivalent to full fine-tuning or is it less powerful than full fine-tuning?**   In Section 3, we showed that prefix-tuning, in principle, has a large capacity to influence the behavior of the model. But then, in this section, we showed that it has some severe limitations, including not being able to affect the attention pattern and only biasing the attention layer activations. These two results seem to be contradicting one another, so how do we reconcile them?

The constructions for the results in Section 3 (described in Appendix A) are simply an algorithm that extracts the completion from a lookup table encoded in the virtual tokens. The attention patterns

---

[4]Likhosherstov et al. (2021) show that a fixed attention head can approximate any sparse attention pattern. However, they require control over all the input embeddings while we can only control the prefix ones.

[5]He et al. (2021) show a similar analysis but do not study the expressiveness of prefix-tuning.

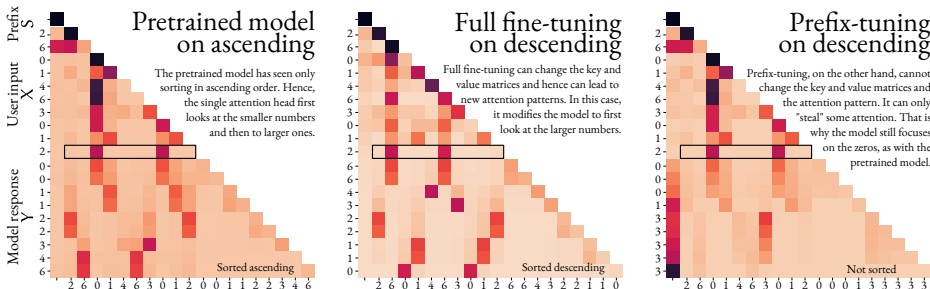

Figure 1: Attention patterns of a small transformer pretrained on sorting in ascending order. The model is given the prefix $S$ and user input $X$ and generates $Y$ autoregressively. We have highlighted the attention when the first response $Y_1$ is being generated. Full fine-tuning sorts in descending order but prefix-tuning cannot as it cannot update the learned attention. Note how the relative attention of $X$ to $X$ in the left and right plots is exactly the same: the prefix cannot change the attention pattern for the same inputs. The relative attention of $X$ to $X$ in the center plot is very different because full fine-tuning can arbitrarily change $\boldsymbol{W}_Q$ and $\boldsymbol{W}_K$.

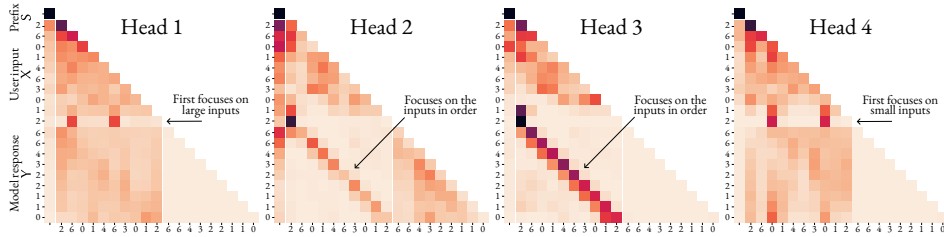

Figure 2: Model pretrained on the four tasks. The four attention heads specialize in the skills necessary to solve these tasks: look at the elements in order, look first at the smallest elements or first at the largest elements.

are simply extracting the current position embedding and the virtual token and hence the attention does not depend on the actual content in the tokens. There is no need to learn a new attention pattern to learn a different map from input to output.[6] Furthermore, the virtual token designates the map precisely by acting as a bias. Therefore, the observations in these two sections do not contradict one another. Soft prompting and prefix-tuning can be on par with full fine-tuning but only in very limited circumstances: when all the knowledge is represented in the virtual token as a lookup table and the model simply extracts the relevant entry. Transformers do not behave like this in practice. Models are typically trained with token inputs rather than virtual tokens. Moreover, if we had a lookup table of the responses to each input we would not need a learning algorithm in the first place.

Therefore, the limitations from this section hold for real-world pretrained transformers. Then how come prefix-tuning has been reported to achieve high accuracy and often to be competitive to full fine-tuning? The next section aims to explain when and why prefix-tuning can work in practice.

## 5 THE BIAS CAN ELICIT SKILLS FROM THE PRETRAINED MODEL

Pretraining potentially exposes a model to different types of completions for the same token sequence. For a string like *I didn't enjoy the movie*, the model may have seen completions such as *I found the acting to be sub par*, *This is negative sentiment* or *Je n'ai pas aimé le film*. Hence, a pretrained model could do text completion, sentiment analysis, or translation. Still, the input does not fully determine the desired completion type and the model can generate any one of them. Hence, following our results from Section 4, we hypothesise that prefix-tuning cannot gain *new knowledge* but can bring to the surface *latent knowledge* present in the pretrained model.[7] We test this hypoth-

---

[6]In a follow-up work (Petrov et al., 2024), we utilize this observation to show that, in fact, there exist pretrained weights for which a transformer can be a universal approximator for sequence-to-sequence functions when prefixed. This is not in contradiction with the present results as these transformers can approximate any function without having to modify their attention mechanism.

[7]A similar hypothesis has also been proposed by Reynolds and McDonell (2021) for fine-tuning in general.

esis by constructing small transformers trained on one or few tasks. We use a minimal transformer model (Karpathy, 2020) to show that prefix-tuning struggles to learn a new task that full fine-tuning can. Then, that prefix-tuning can easily elicit a latent skill from pretraining. Finally, we show how it can even learn *some* new tasks, provided they can be solved by combining pretraining skills.

**Prefix-tuning may not learn a new task requiring a different attention pattern.** To check if prefix-tuning can learn a new task, we train a 1-layer, 1-head transformer to sort numbers into ascending order and then fine-tune it to sort in descending order. During training, the model sees random sequences of 10 digits from 0 to 7 followed by their ascending sorted order. The pretrained accuracy (fully matching the sorted sequence) is 91%. Full fine-tuning on the descending task leads to

Table 1: A transformer pretrained on sorting in ascending order cannot be prefix-tuned to sort in descending order. 10 random seeds.

|                        | Ascending | Descending |
| ---------------------- | --------- | ---------- |
| Pretrain on asc.       | 91±5%     | 0±0%       |
| Full fine-tune on desc.| 0±0%      | 85±5%      |
| Prefix-tune on desc.   | 0±0%      | 0±0%       |

85% test accuracy, hence full fine-tuning successfully learns the new task. However, prefix-tuning with a prefix size $n_S$=1 results in 0% accuracy, hence prefix-tuning fails to learn the new task at all.

The attention patterns in Figure 1 show why this is the case: the pretrained model learns to attend first to the smallest numbers and then to the larger ones. When fully fine-tuned, the attention patterns are reversed: they now first attend to the largest values. However, following Section 4, prefix-tuning cannot change the attention pattern over the input sequence and will still attend to the smallest values. Hence, prefix-tuning may indeed struggle to learn a new task requiring new attention patterns.

**Prefix-tuning can elicit a skill from the pretrained model.** The second part of our hypothesis was that prefix-tuning can elicit latent skills in the pretrained model. To test that, we pretrain a 1-layer, 4-head model with solutions sorted in ascending ($\nearrow$) or descending ($\searrow$) order, or adding one (+1) or two (+2) to each element of the input sequence. Each solution is shown with 25% probability. The model has no indication of what the task is, hence, it as-

Table 2: A transformer pretrained on several tasks can be prefix-tuned for one of them. 10 random seeds.

| Accuracy on:              | $\nearrow$ | $\searrow$ | +1      | +2      |
| ------------------------- | ---------- | ---------- | ------- | ------- |
| Pretrained                | 25±13%     | 25±12%     | 24±11%  | 22±7%   |
| Prefix-tune on $\nearrow$ | 95± 2%     | 0± 0%      | 0± 0%   | 0±0%    |
| Prefix-tune on $\searrow$ | 0± 0%      | 90± 3%     | 1± 1%   | 1±1%    |
| Prefix-tune on +1         | 0± 0%      | 1± 3%      | 95± 6%  | 0±1%    |
| Prefix-tune on +2         | 0± 0%      | 0± 0%      | 1± 2%   | 98±5%   |

signs equal probability to all tasks, as shown in the first row in Table 2. Full fine-tuning for each task naturally results in high accuracy. However, prefix-tuning ($n_S$=1) can also reach accuracy above 90% for all tasks. Compared to the previous case, prefix-tuning is more successful here because the pretrained model contains the attention mechanisms for solving the four tasks, as shown in Figure 2.

If all a prefix does is bias the attention layer activations, how can it steer the model to collapse its distribution onto one task? This is likely due to the attention block solving all tasks in parallel and placing their solutions in different subspaces of the residual stream (intermediate representation, Elhage et al., 2021). As the MLP needs to select one solution to generate, a further indicator on the selected task (or lack of selection thereof) should also be represented. The bias induced by the prefix then acts on this "selection subspace" to nudge the MLP to select the desired solution.

This can be clearly seen from the activations of the attention layer at the last input position ($X_{n_X}$): the position where the task selection happens as the first output element fully describes the task. Figure 4 shows plots of randomly selected dimensions of the residual stream with and without a prefix. The attention block activations of the pretrained model (without prefix) show no correlation with the output it is about to generate, demonstrating that the choice of completion is indeed not determined by the attention block. However, the prefix-tuned activations for the same inputs are clustered as a result of the prefix-induced bias. This indicates that the bias induced by the prefix may act as a "task selector" of the subspace of the residual stream specializing in the desired task.

**Prefix-tuning can combine knowledge from pretraining tasks to solve new tasks.** Prefix-tuning eliciting one type of completion learned in pretraining starts to explain its practical utility. Still, prefix-tuning seems to be successful also at tasks that the pretrained model has not seen. As we showed above, a model trained to sort in one order cannot be prefix-tuned to sort in the other. Then how is it possible for prefix-tuning to learn a new task? We posit that this can happen, as long as the "skill" required to solve the new task is a combination of "skills" the pretrained model has seen.

We test this by pretraining a 40-layer 4-head model with the same four tasks. We prefix-tune ($n_S$=12) for two new tasks: incrementing the ascending sorted sequence ($\nearrow$+1) and double histogram (mapping each element to the number of elements with the same value, e.g., 3,0,0,1↦1,2,2,1, $\mathbb{H}$). The pretrained model has not seen either task. Prefix-tuning results in 93% accuracy for $\nearrow$+1 which is a combination of the $\nearrow$ and +1 pretraining tasks and just 1% for the $\mathbb{H}$ task which requires different skills: finding other instances

Table 3: Prefix tuning can learn a new task requiring only pretraining skills ($\nearrow$+1) but cannot learn a completely new task ($\mathbb{H}$). Average accuracy over 3 seeds.

| Accuracy on: | $\nearrow$ | $\searrow$ | +1 | +2 | $\nearrow$+1 | $\mathbb{H}$ |
|---|---|---|---|---|---|---|
| Pretrained | 17% | 23% | 34% | 25% | 0% | 0% |
| Prefix-tune on $\nearrow$ | 100% | 0% | 0% | 0% | 0% | 0% |
| Prefix-tune on $\searrow$ | 0% | 100% | 0% | 0% | 0% | 0% |
| Prefix-tune on +1 | 0% | 0% | 100% | 0% | 0% | 0% |
| Prefix-tune on +2 | 0% | 0% | 0% | 100% | 0% | 0% |
| Prefix-tune on $\nearrow$+1 | 0% | 0% | 0% | 0% | 93% | 0% |
| Prefix-tune on $\mathbb{H}$ | 0% | 0% | 0% | 0% | 0% | 1% |

of the same token and counting. $\mathbb{H}$ is not a hard task: it requires 2 layers and 2 heads to be solved exactly (Weiss et al., 2021). Therefore, prefix-tuning is can indeed combine skills that the model has learned in order to solve a novel task but may not learn a completely new task requiring new skills.

# 6 EFFECTS OF PREFIX-TUNING BEYOND THE SINGLE ATTENTION LAYER

Section 4 focused exclusively on a single attention layer. Still, even if a prefix only induces a bias on its output, this bias can exhibit complex behaviors via the subsequent MLPs and attention layers. This section shows how a prefix can change the attention pattern of the following attention layer but only in a linear fashion while full fine-tuning also has bilinear effects. Appendix C further argues that the representational capacity of prefix-tuning may be limited. Therefore, prefix-tuning appears to be less expressive than full fine-tuning, even with the same number of learnable parameters.

**Prefix-tuning can change the attention, albeit the one of the next layer**   Let us examine how the prefix of one attention layer affects the following one. Assume no MLPs, residual connections or layer norms: the output $t_i^{(1)}$ of the first is the input $x_i^{(2)}$ of the second. The pretrained outputs are $t_i^{(1)} = \sum_{j=1}^p A_{ij}^{(1)} W_V^{(1)} x_j^{(1)}$, resulting in the second layer attention $\tilde{A}_{ij}^{(2)} = T/\sqrt{k}\, t_i^{(1)\top} H^{(2)} t_j^{(1)}$. Here $\tilde{A}_{ij}$ is the pre-softmax attention, i.e., $A_{ij} = \exp \tilde{A}_{ij} / \sum_{r=1}^p \exp \tilde{A}_{ir}$. For prefix-tuning we then have:

$$t_i^{\mathrm{pt}(1)} = A_{i0}^{\mathrm{pt}(1)} W_V s_1^{(1)} + \sum_{j=1}^p A_{ij}^{\mathrm{pt}(1)} W_V^{(1)} x_j^{(1)} \overset{(7)}{=} \underbrace{A_{i0}^{\mathrm{pt}(1)}}_{\alpha_i} \underbrace{W_V s_1^{(1)}}_{\mu} + (1 - A_{i0}^{\mathrm{pt}(1)}) t_i^{(1)},$$

$$\tilde{A}_{ij}^{\mathrm{pt}(2)} = \tfrac{T}{\sqrt{k}}\, t_i^{\mathrm{pt}(1)\top} H^{(2)} t_j^{\mathrm{pt}(1)},$$
$$= \tfrac{T}{\sqrt{k}} (\alpha_i \alpha_j \underbrace{\mu^\top H^{(2)} \mu}_{\text{constant}} + \alpha_j (1-\alpha_i) \underbrace{t_i^{(1)\top} H^{(2)} \mu}_{\text{depends only on } t_i^{(1)}} + \alpha_i (1-\alpha_j) \underbrace{\mu^\top H^{(2)} t_j^{(1)}}_{\text{depends only on } t_j^{(1)}} + (1-\alpha_i)(1-\alpha_j) \underbrace{t_i^{(1)\top} H^{(2)} t_j^{(1)}}_{\text{pretrained attention } \tilde{A}_{ij}^{(2)}}).$$

The presence of $\mu$ shows that the prefix of layer 1 can change the attention pattern of the following layer. This change is content-specific: the second and the third terms depend on the inputs, hence a simple bias can affect the attention when passed through MLPs and further attention blocks. Compare with Equation (6), which showed a prefix cannot change the attention of the same layer. Still, even considering this cross-layer effect, prefix-tuning is more limited in its expressiveness than full fine-tuning. While the second and the third terms are input-dependent, each depends on one input position only. The prefix does not change the bilinear dependency on both the query and key. This is something that the full fine-tuning can achieve: $\tilde{A}_{ij}^{\mathrm{ft}(2)} = T/\sqrt{k}\, t_i^{\mathrm{ft}(1)\top} (H^{(2)} + \Delta H^{(2)}) t_j^{\mathrm{ft}(1)}$.

**Even if prefix-tuning could be a universal approximator, it would not be parameter-efficient.** Prefix-tuning appears to be less parameter-efficient than other comparable approaches. We designed an experiment to this end. Our pretrained model in Section 5 failed to learn the double histogram task ($\mathbb{H}$). A rank-1 Low Rank Adaptation (LoRA, Hu et al., 2021) applied only to the MLPs in a 4-layer 4-head model pretrained in the exact same way results in 92% accuracy on the $\mathbb{H}$ task. The number of parameters for the LoRA fine-tuning is exactly the same as for a prefix of size 12. However, as can be expected from the results in Section 5, training this prefix results in 0% accuracy. Hence, prefix-tuning fails at a task that LoRA with the same number of parameters can learn.

## 7 DISCUSSION AND RELATED WORKS

**Understanding fine-tuning and prefix-tuning.** Prior works show that prefixes have low intrinsic dimension allowing transfer to similar tasks and initialization of prefixes for new tasks (Qin et al., 2021; Su et al., 2022; Zhong et al., 2022; Wang et al., 2022b; Zheng et al., 2023). In this work, we offered theoretical insights into their results: this subspace is the span of the prefix-induced bias. Another line of work shows that skills can be localized in the parameter space of pretrained models (Wang et al., 2022a; Panigrahi et al., 2023). Here, we showed that it is also possible to identify subspaces of the residual stream corresponding to individual tasks and select them via prefix-tuning.

**Prompting and in-context learning.** Prompting and in-context learning are a special case of prefix-tuning. Therefore, the limitations and mechanisms discussed in this work apply to prompting as well: prompts cannot change the distribution of attention of the first attention layer over the content following it and can only induce a bias on the output of this layer (Section 4). Even considering the cross-layer effects, a prompt is strictly less expressive than full fine-tuning (Section 6) and prompting is unlikely to enable the model to solve a completely new task. Our theory thus explains why Kossen et al. (2023) observed that in-context examples cannot overcome pre-training skills.

While context-based fine-tuning approaches may not learn *arbitrary* new tasks, as shown in Section 5, they can leverage pre-trained skills. Wies et al. (2023) have PAC-learnability results that also show that when pretraining is on a mixture tasks, they can be efficiently learned via in-context learning, Moreover, transformers can learn linear models in-context by mimicking gradient descent (Von Oswald et al., 2023) or approximating matrix inversion (Akyürek et al., 2022). This is consistent with our theory: the prediction updates are enacted as biases in the attention block activations. Hence, despite the limitations discussed in this work, context-based methods can result in powerful fine-tuning if the pretrained model has "transferable skills" such as algorithmic fundamentals. Still, in-context learning will likely fail for non-algorithmic tasks, e.g., translating to a language that the model has never seen before, even if large number of translation pairs are provided in-context.

**Implications for model interpretability.** An open question for language model interpretability is whether attention is sufficient for explainability (Jain and Wallace, 2019; Wiegreffe and Pinter, 2019). Section 5 points toward the negative: by interfering in the *output* of the attention layer with the bias induced by a prefix, we can change the behavior of the model, without changing its attention. On the flip side, prefix-tuning can be used to understand what "skills" a model has: if prefix-tuning for a task fails, then the model likely lacks one of the key "skills" for that task.

**Limitations.** The present analysis is largely limited to prefixing with prompts, soft prompts and for prefix-tuning. While our theoretical results hold for suffix-tuning, they do not necessarily apply to suffixing with prompts or soft prompts. That is because the deeper representations for prompt and soft prompt suffixes would depend on the previous positions. This does not apply to suffix-tuning as it fixes all intermediate representations. Therefore, whether suffixing is more expressive than prefixing remains an open question. Separately, while we provided evidence towards context-based fine-tuning methods being parameter inefficient learners, the formal analysis of the conditions under which they may be universal approximators remain an open question. Finally, we mostly considered simple toy problems. In practice, however, language models are pretrained with very large datasets and can pick up very complex behaviors. Hence, the extent to which the limitations we demonstrated apply to large-scale pretrained transformers also remains for future work.

## 8 CONCLUSION

This paper formally showed that fine-tuning techniques working in embedding space, such as soft prompting and prefix-tuning, are strictly more expressive than prompting which operates in the discrete token space. However, we then demonstrated that despite this larger expressivity, prefix-tuning suffers from structural limitations that prevent it from learning new attention patterns. As a result, it can only bias the output of the attention layer in a direction from a subspace of rank equal to the size of the prefix. We showed that this results in practical limitations by constructing minimal transformers where prefix tuning fails to solve a simple task. This result seems to be at odds with the empirical success of prefix-tuning. We provided explanations towards that. First, we showed that prefix-tuning can easily elicit a skill the pretrained model already has and can even learn a new task, if it has picked up the skills to solve it during pretraining. Second, we showed that the effect of the prefix-induced bias is more complicated and powerful when combined with downstream non-linear operations. However, it appears to be still less expressive than full fine-tuning.

## REPRODUCIBILITY STATEMENT

In order to facilitate the reproduction of our empirical results, validating our theoretical results, and further studying the properties of context-based fine-tuning, we release all our code and resources used in this work. Furthermore, in Appendix A we offer explicit constructions of transformers with the properties discussed in Section 3. We also provide Python implementations of these constructions that validate their correctness.

## ACKNOWLEDGEMENTS

This work is supported by a UKRI grant Turing AI Fellowship (EP/W002981/1) and the EPSRC Centre for Doctoral Training in Autonomous Intelligent Machines and Systems (EP/S024050/1). AB has received funding from the Amazon Research Awards. We also thank the Royal Academy of Engineering and FiveAI.

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

# A  Constructing transformers that utilize the capacity of the embedding space

## A.1  Unconditional generation for a single virtual token

This section provides an explicit construction of a transformer with the properties described in Theorem 1. The goal is to construct a transformer that, by varying the choice of the virtual token, can generate any sequence of $N$ tokens.

First, we need to specify how we encode the target sequence $(\textsc{y}_1, \ldots, \textsc{y}_N)$ into the virtual token $\boldsymbol{s}_1$. We chose the size of the embedding (and hence of $\boldsymbol{s}_1$) to be $N$. This way, each element of $\boldsymbol{s}_1$ can represent one position of the target sequence. We then represent the token value by discretizing each element of $\boldsymbol{s}_1$ into $V$ levels:

$$\boldsymbol{s}_1 = \left((\textsc{y}_1 - 1)/V, \ldots, (\textsc{y}_N - 1)/V\right).$$

Note that this means that each element of $\boldsymbol{s}_1$ is in $[0, 1)$.

When predicting the token for the $i + 1$ position, the transformer needs to pick the $i$-th element of $\boldsymbol{s}_1$, and then decode the corresponding value as a one-hot encoding representing the $\textsc{y}_i$-th token.

We extract the $i$-th element of $\boldsymbol{s}_1$ using one attention block of two heads. The `fst` head always looks at the first position which is our virtual token $\boldsymbol{s}_1$. For that purpose we create an attention head that always has $\boldsymbol{A}_{ij}^{\mathrm{fst}} = 1$ if $j = 1$ and $\boldsymbol{A}_{ij}^{\mathrm{fst}} = 0$ otherwise together with a value matrix $\boldsymbol{W}_V^{\mathrm{fst}}$ that extracts the embedding. This is achieved with

$$\boldsymbol{W}_Q^{\mathrm{fst}} = [\boldsymbol{0}_N, \boldsymbol{1}_N], \qquad \boldsymbol{W}_K^{\mathrm{fst}} = [\boldsymbol{0}_N, 1, \boldsymbol{1}_{N-1}], \qquad \boldsymbol{W}_V^{\mathrm{fst}} = [\boldsymbol{I}_N, \boldsymbol{0}_{N \times N}], \qquad (8)$$

and a sufficiently high inverse temperature parameter $T$.

The `pos` head instead extracts the one-hot encoding of the current position. This can be done with an attention head that always attends only to the current position and a value matrix $\boldsymbol{W}_V^{\mathrm{pos}}$ that extracts the position embedding as a one-hot vector:

$$\boldsymbol{W}_Q^{\mathrm{pos}} = [\boldsymbol{0}_{N \times N}, \boldsymbol{I}_N], \qquad \boldsymbol{W}_K^{\mathrm{pos}} = [\boldsymbol{0}_{N \times N}, \boldsymbol{I}_N], \qquad \boldsymbol{W}_V^{\mathrm{pos}} = [\boldsymbol{0}_{N \times N}, \boldsymbol{I}_N]. \qquad (9)$$

When the outputs of these two attention heads are summed, then only the element of $\boldsymbol{s}_1$ that corresponds to the current position will be larger than 1. From Equation (2) the output at the $i$-th position of the attention block is:

$$\boldsymbol{t}_i = \sum_{j=1}^{p} \boldsymbol{A}_{ij}^{\mathrm{fst}} \boldsymbol{x}_j + \sum_{j=1}^{p} \boldsymbol{A}_{ij}^{\mathrm{pos}} \boldsymbol{e}_N(j) = \boldsymbol{s}_1 + \boldsymbol{e}_N(i),$$

where $\boldsymbol{x}_1 = \boldsymbol{s}_1$ and $\boldsymbol{x}_j = E_{:,\textsc{y}_{j-1}}$ for $j > 1$.

We can extract the value of $\boldsymbol{s}_1$ corresponding to the current position by substracting 1 from the hidden state and apply ReLU: $\hat{\mathcal{L}}_{\mathrm{ex}} = \hat{\mathcal{L}}[\boldsymbol{I}_N, -\boldsymbol{1}_N]$. Now, we are left with only one non-zero entry and that's the one corresponding to the next token. We can retain only the non-zero entry if we just sum all the entries of the hidden state with $\hat{\mathcal{L}}_{\mathrm{sum}} = \hat{\mathcal{L}}[\boldsymbol{1}_N^\top, 0]$.

The final step is to map this scalar to a $V$-dimensional vector which has its maximum value at index $\textsc{y}_i$. This task is equivalent to designing $V$ linear functions, each attaining its maximum at one of $0, 1/V, \ldots, (V-1)/V$. To construct this, we use the property of convex functions that their tangent is always under the plot of the function. Therefore, given a convex function $\gamma(x)$, we construct the $i$-th linear function to be simply the tangent of $\gamma$ at $i-1/V$. If we take $\gamma(x) = (x - 1/2)^2$, this results in the following linear layer:

$$\mathcal{L}_{\mathrm{proj}} = \mathcal{L}\left[\left[\frac{2(1-1)}{V} - 1, \ldots, \frac{2(V-1)}{V} - 1\right]^T, \left[\frac{1}{4} - \frac{(1-1)^2}{V^2}, \ldots, \frac{1}{4} - \frac{(V-1)^2}{V^2}\right]^\top\right]. \qquad (10)$$

Figure 3 shows the predictors for each individual token id.

With just two attention heads and three linear layers, the transformer $\mathcal{A}[(\boldsymbol{W}_Q^{\mathrm{fst}}, \boldsymbol{W}_Q^{\mathrm{pos}}), (\boldsymbol{W}_K^{\mathrm{fst}}, \boldsymbol{W}_K^{\mathrm{pos}}), (\boldsymbol{W}_V^{\mathrm{fst}}, \boldsymbol{W}_V^{\mathrm{pos}})] \, \mathring{,} \, \hat{\mathcal{L}}_{\mathrm{ex}} \, \mathring{,} \, \hat{\mathcal{L}}_{\mathrm{sum}} \, \mathring{,} \, \mathcal{L}_{\mathrm{proj}} \, \mathring{,} \, \mathrm{softmax}$ achieves the upper bound of $V^N$ unique outputs by controlling a single virtual token at its input. Note that for this

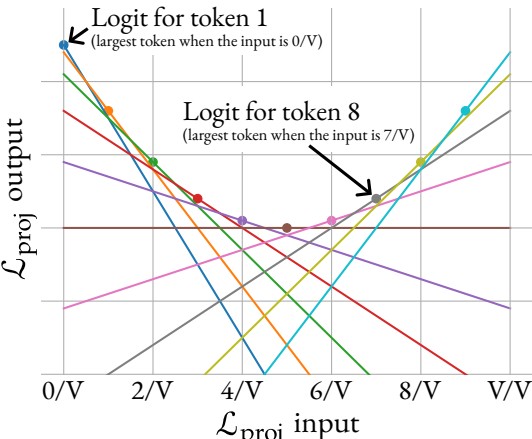

Figure 3: Illustration of the predictors for each token in the $\mathcal{L}_{\text{proj}}$ linear layer for $V = 10$. The layer is constructed in such a way that the $i$-th token has the highest confidence when the input is $^{i-1}/V$.

construction, the choice of embedding matrix $E \in \mathbb{R}^{N \times V}$ does not matter. The same transformer architecture can generate only $V$ unique outputs if we only control the first token instead. Therefore, it is indeed the case that the embedding space has exponentially more capacity for control than the token space. You can see this transformer implemented and running in practice in Section 2 of this notebook.

## A.2 CONDITIONAL GENERATION FOR A SINGLE VIRTUAL TOKEN ($n_X = n_Y = 1$)

This section provides an explicit construction of a transformer with the properties described in Theorem 2. The goal is to construct a transformer that, by varying the choice of the virtual token, can cause the model to act as any map $m : [1, \ldots, V] \to [1, \ldots, V]$. In other words, by selecting the virtual token, we can fully control how the model will respond to any token the user may provide.

First, we need to specify how the map $m$ will be encoded in the virtual token $s_1$. We choose the embedding size $d_e$ to be $V$. Now, we can use the same encoding scheme as before, but now each element in $s_1$ corresponds to a different user token, rather than to a position in the generated sequence:

$$s_1 = \left( {m(1)}/{V}, \ldots, {m(V)}/{V} \right).$$

Therefore, the first element of $s_1$ designates the response if the user provides token 1, the second element is the response to the token 2, and so on.

Extracting the $\mathrm{Y}_i$-th value from $s_1$ and decoding it can be done in a very similar way as for the unconditional case. The only difference is that instead of looking at the user input position, we look at its value. Take $E = I_V$ and $N = 2$.

Hence we have the following $\mathtt{val}$ head (only differing in the $W_V$ matrix from Equation (9)):

$$W_Q^{\text{val}} = [\mathbf{0}_{2 \times V}, I_2], \qquad W_K^{\text{val}} = [\mathbf{0}_{2 \times V}, I_2], \qquad W_V^{\text{val}} = [I_V, \mathbf{0}_{V \times 2}].$$

We also need embedding of the first token, so we have a modified version of Equation (8):

$$W_Q^{\text{fst}} = [\mathbf{0}_V, 1, 1], \qquad W_K^{\text{fst}} = [\mathbf{0}_V, 1, 0], \qquad W_V^{\text{fst}} = [I_V, \mathbf{0}_{V \times 2}].$$

And hence the output of this attention block at the second position would be:

$$t_2 = \sum_{j=1}^{2} A_{ij}^{\text{fst}} x_j + \sum_{j=1}^{2} A_{ij}^{\text{val}} W_V^{\text{fst}} x_j = s_1 + e_V(\mathrm{Y}_1).$$

Similarly to the unconditional case, only the entry of $t_2$ corresponding to the user token will have a value above 1 and that value would be $1 + m(x_1)/V$.

We can now extract the one-hot representation of the target token using the same approach as before, just adjusting for the different hidden state size: $\hat{\mathcal{L}}_{\text{ex}} = \hat{\mathcal{L}}[\boldsymbol{I}_V, -\boldsymbol{1}_V]$, $\hat{\mathcal{L}}_{\text{sum}} = \hat{\mathcal{L}}[\boldsymbol{1}_V^\top, 0]$, and the same projection had as before (Equation (10)). The final transformer is then: $\mathcal{A}[(\boldsymbol{W}_Q^{\text{fst}}, \boldsymbol{W}_Q^{\text{val}}), (\boldsymbol{W}_K^{\text{fst}}, \boldsymbol{W}_K^{\text{val}}), (\boldsymbol{W}_V^{\text{fst}}, \boldsymbol{W}_V^{\text{val}})] \, \mathring{,} \, \hat{\mathcal{L}}_{\text{ex}} \, \mathring{,} \, \hat{\mathcal{L}}_{\text{sum}} \, \mathring{,} \, \mathcal{L}_{\text{proj}} \, \mathring{,} \, \text{softmax}$. You can see this transformer implemented and running in practice in Section 3 here.

### A.3 CONDITIONAL GENERATION FOR LONGER RESPONSES ($n_X = 1, n_Y > 1$)

We can obtain longer responses via a simple extension. If the response length is $N_0$, then we can encode the map $m : [1, \ldots, V] \to [1, \ldots, V]^{N_0}$ in $N_0$ virtual tokens, each corresponding to one of the target positions:

$$\boldsymbol{s}_i = \left(m(1)_i/V, \ldots, m(V)_i/V\right) \text{ for } i = 1, \ldots, N_0.$$

For this model we would then have $N = 2N_0$ and $d_e = V$.

First, we need a head that always looks at the token provided by the user, which will be at position $N_o + 1$:

$$\boldsymbol{W}_Q^{\text{user}} = [\boldsymbol{0}_V, \boldsymbol{1}_N], \qquad \boldsymbol{W}_K^{\text{user}} = [\boldsymbol{0}_{(V+N_o)}, 1, \boldsymbol{0}_{(N_o-1)}], \qquad \boldsymbol{W}_V^{\text{user}} = [\boldsymbol{I}_V, \boldsymbol{0}_{V \times N}].$$

In order to consume the map at the right location, we need to also look at the embedding of the token $N_o$ positions before the one we are trying to generate:

$$\boldsymbol{W}_Q^{\text{back}} = \begin{bmatrix} \boldsymbol{0}_{N_0 \times (N_0+V)} & \boldsymbol{I}_{N_0} \\ \boldsymbol{0}_{N_0 \times (N_0+V)} & \boldsymbol{0}_{N_0 \times N_0} \end{bmatrix}, \qquad \boldsymbol{W}_K^{\text{back}} = [\boldsymbol{0}_{N \times V}, \boldsymbol{I}_N], \qquad \boldsymbol{W}_V^{\text{back}} = [\boldsymbol{I}_V, \boldsymbol{0}_{V \times N}].$$

From here on, the decoding is exactly the same as in the $n_X = n_Y = 1$ case. The final transformer is then: $\mathcal{A}[(\boldsymbol{W}_Q^{\text{user}}, \boldsymbol{W}_Q^{\text{back}}), (\boldsymbol{W}_K^{\text{user}}, \boldsymbol{W}_K^{\text{back}}), (\boldsymbol{W}_V^{\text{user}}, \boldsymbol{W}_V^{\text{back}})] \, \mathring{,} \, \hat{\mathcal{L}}_{\text{ex}} \, \mathring{,} \, \hat{\mathcal{L}}_{\text{sum}} \, \mathring{,} \, \mathcal{L}_{\text{proj}} \, \mathring{,} \, \text{softmax}$. You can see this transformer implemented and running in practice in Section 4 here.

### A.4 CONDITIONAL GENERATION FOR LONGER USER INPUTS ($n_X > 1, n_Y = 1$)

Finally, we consider the case when the user input X is longer. This is a bit more complicated because we need to search through a domain of size $V^V$. We will only consider the case with $n_X = 2$ where we would need two attention layers. A similar approach can be used to construct deeper models for $n_X > 2$. Finally, combining the strategy in the previous section for longer responses with the strategy in this section for longer user inputs allows us to construct transformers that map from arbitrary length user strings to arbitrary length responses.

In order to encode a map $m : [1, \ldots, V]^2 \to [1, \ldots, V]$ into a single virtual token we would need a more involved construction than before. Similarly to how we discretized each element of the virtual token $\boldsymbol{s}_1$ in $V$ levels before, we are going to now discretize it into $V^V$ levels. Each one of these levels would be one of the $V^V$ possible maps from the *second* user token to the response. The first user token would be used to select the corresponding element of $\boldsymbol{s}_1$. Then this scalar will be "unpacked" into a new vector of $V$ elements using the first attention block. Then, the second user token will select an element from this unpacked vector, which will correspond to the target token.

We construct the virtual token as follows:

$$\boldsymbol{s}_1 = \left[\sum_{i=1}^V m_1(i) \times \frac{V^{i-1}}{V^V}, \ldots, \sum_{i=1}^V m_V(i) \times \frac{V^{i-1}}{V^V}\right],$$

where $m_f(x) = m(f, x)$ is a map from the second user token to the response when the first token is fixed to be $f$.

An additional change from the previous constructions is that we are going to divide the residual stream into two sections. This is in line with the theory that different parts of the residual stream specialize for different communications needs by different attention heads (Elhage et al., 2021). We will use the first half of the residual stream to extract and "unpack" the correct mapping from second token to target token, while the second half of the residual stream will be used to copy the second token value so that the second attention layer can use it to extract the target. As usual, the embedding

matrix will be the identity matrix: $\boldsymbol{E} = \boldsymbol{I}_V$. Finally, for convenience, we will also use a dummy zero virtual token that we will attend to when we want to not attend to anything. This results in context size $N = 4$ with the input being

$$\left( \begin{bmatrix} 0_V \\ \boldsymbol{e}_N(1) \end{bmatrix}, \begin{bmatrix} \boldsymbol{s}_1 \\ \boldsymbol{e}_N(2) \end{bmatrix}, \begin{bmatrix} \boldsymbol{E}_{:,X_1} \\ \boldsymbol{e}_N(3) \end{bmatrix}, \begin{bmatrix} \boldsymbol{E}_{:,X_2} \\ \boldsymbol{e}_N(4) \end{bmatrix} \right) = \left( \begin{bmatrix} \boldsymbol{0}_V \\ \boldsymbol{e}_N(1) \end{bmatrix}, \begin{bmatrix} \boldsymbol{s}_1 \\ \boldsymbol{e}_N(2) \end{bmatrix}, \begin{bmatrix} \boldsymbol{e}_V(X_1) \\ \boldsymbol{e}_N(3) \end{bmatrix}, \begin{bmatrix} \boldsymbol{e}_V(X_2) \\ \boldsymbol{e}_N(4) \end{bmatrix} \right).$$

We want the output at the last position to be the target $m(X_1, X_2)$, that is:

$$\arg\max_{u \in 1, \ldots, V} \boldsymbol{y}_{4,u} = m(X_1, X_2) \text{ for any } m, X_1, X_2.$$

The first attention block will have three attention heads.

As before, we want to extract the value of $\boldsymbol{s}_1$ that corresponds to the first token the user provided $(X_1)$ and place it in the first half of the residual stream. We want only the third position to do that, while the rest of the positions keep the first half of their residual stream with zeros. Hence we have the following `fst` head:

$$\boldsymbol{W}_Q^{\text{fst}} = \begin{bmatrix} \boldsymbol{0}_{2 \times V} & \begin{matrix} 1 & 1 & 0 & 1 \\ 0 & 0 & 1 & 0 \end{matrix} \end{bmatrix}, \quad \boldsymbol{W}_K^{\text{fst}} = \begin{bmatrix} \boldsymbol{0}_{2 \times V} & \begin{matrix} 1 & 0 & 0 & 0 \\ 0 & 1 & 0 & 0 \end{matrix} \end{bmatrix}, \quad \boldsymbol{W}_V^{\text{fst}} = \begin{bmatrix} \boldsymbol{I}_V & \boldsymbol{0}_{V \times N} \\ \boldsymbol{0}_{V \times V} & \boldsymbol{0}_{V \times N} \end{bmatrix}.$$

The `user1` head extracts the value of the first user-provided token $(X_1)$ and also places it in the first half of the residual stream:

$$\boldsymbol{W}_Q^{\text{user1}} = \begin{bmatrix} \boldsymbol{0}_{2 \times V} & \begin{matrix} 1 & 1 & 0 & 1 \\ 0 & 0 & 1 & 0 \end{matrix} \end{bmatrix}, \quad \boldsymbol{W}_K^{\text{user1}} = \begin{bmatrix} \boldsymbol{0}_{2 \times V} & \begin{matrix} 1 & 0 & 0 & 0 \\ 0 & 0 & 1 & 0 \end{matrix} \end{bmatrix}, \quad \boldsymbol{W}_V^{\text{user1}} = \begin{bmatrix} \boldsymbol{I}_V & \boldsymbol{0}_{V \times N} \\ \boldsymbol{0}_{V \times V} & \boldsymbol{0}_{V \times N} \end{bmatrix}.$$

And the `user2` head does the same for the value of the second user-provided token $(X_2)$, placing it in the second half of the residual stream:

$$\boldsymbol{W}_Q^{\text{user2}} = \begin{bmatrix} \boldsymbol{0}_{2 \times V} & \begin{matrix} 1 & 1 & 1 & 0 \\ 0 & 0 & 0 & 1 \end{matrix} \end{bmatrix}, \quad \boldsymbol{W}_K^{\text{user2}} = \begin{bmatrix} \boldsymbol{0}_{2 \times V} & \begin{matrix} 1 & 0 & 0 & 0 \\ 0 & 0 & 0 & 1 \end{matrix} \end{bmatrix}, \quad \boldsymbol{W}_V^{\text{user2}} = \begin{bmatrix} \boldsymbol{0}_{V \times V} & \boldsymbol{0}_{V \times N} \\ 2\boldsymbol{I}_V & \boldsymbol{0}_{V \times N} \end{bmatrix},$$

where the factor 2 is there because, as usual, the first linear layer will subtract 1 from everything in order to extract the value selected by the first token.

This linear layer looks as usual: $\hat{\mathcal{L}}_{\text{ex2}} = \hat{\mathcal{L}}[\boldsymbol{I}_{2V}, -\boldsymbol{1}_{2V}]$. The result is that the first $V$ elements will be 0 except one which designates which map from second user token to output we should use, and the second $V$ elements have a one hot-encoding of the second user token. Constructing an MLP that unpacks the mapping can become quite involved so we do not provide an explicit form for it. But from the universal approximation theorems and the finiteness of the domain and range, we know that such an MLP should exist. We thus designate by `unpack` the MLP that decodes the first half of the residual stream to:

$$\left( \frac{m_{X_1}(1)}{V}, \ldots, \frac{m_{X_1}(V)}{V} \right)$$

and keeps the second half unchanged.

And now, by using two attention heads, the second attention block extracts the value of the above vector at the position designated by the second token, in a fashion not dissimilar to all the previous cases:

$$\boldsymbol{W}_Q^{\text{emb}} = [\boldsymbol{0}_V^{\top}, \boldsymbol{1}_V^{\top}], \qquad \boldsymbol{W}_K^{\text{emb}} = [\boldsymbol{1}_V^{\top}, \boldsymbol{0}_V^{\top}], \qquad \boldsymbol{W}_V^{\text{emb}} = [\boldsymbol{I}_V \quad \boldsymbol{0}_{V \times V}],$$

$$\boldsymbol{W}_Q^{\text{user2'}} = [\boldsymbol{0}_V^{\top}, \boldsymbol{1}_V^{\top}], \qquad \boldsymbol{W}_K^{\text{user2'}} = [\boldsymbol{0}_V^{\top}, \boldsymbol{1}_V^{\top}], \qquad \boldsymbol{W}_V^{\text{user2'}} = [\boldsymbol{0}_{V \times V} \quad \boldsymbol{I}_V],$$

And finally, with $\hat{\mathcal{L}}_{\text{ex}} = \hat{\mathcal{L}}[\boldsymbol{I}_V, -\boldsymbol{1}_V]$, $\hat{\mathcal{L}}_{\text{sum}} = \hat{\mathcal{L}}[\boldsymbol{1}_V^{\top}, 0]$, and the same projection had as before (Equation (10)), we get the target token. The final transformer is then: $\mathcal{A}[(\boldsymbol{W}_Q^{\text{fst}}, \boldsymbol{W}_Q^{\text{user1}}, \boldsymbol{W}_Q^{\text{user2}}), (\boldsymbol{W}_K^{\text{fst}}, \boldsymbol{W}_K^{\text{user1}}, \boldsymbol{W}_K^{\text{user2}}), (\boldsymbol{W}_V^{\text{fst}}, \boldsymbol{W}_V^{\text{user1}}, \boldsymbol{W}_V^{\text{user2}})] \mathbin{\text{\scriptsize 9}} \hat{\mathcal{L}}_{\text{ex2}} \mathbin{\text{\scriptsize 9}} \texttt{unpack} \mathbin{\text{\scriptsize 9}}$ $\mathcal{A}[(\boldsymbol{W}_Q^{\text{emb}}, \boldsymbol{W}_Q^{\text{user2'}}), (\boldsymbol{W}_K^{\text{emb}}, \boldsymbol{W}_K^{\text{user2'}}), (\boldsymbol{W}_V^{\text{emb}}, \boldsymbol{W}_V^{\text{user2'}})] \mathbin{\text{\scriptsize 9}} \hat{\mathcal{L}}_{\text{ex}} \mathbin{\text{\scriptsize 9}} \hat{\mathcal{L}}_{\text{sum}} \mathbin{\text{\scriptsize 9}} \mathcal{L}_{\text{proj}} \mathbin{\text{\scriptsize 9}} \text{softmax}$. You can see this transformer implemented and running in practice in Section 5 here.

Figure 4: Attention block activations for ten sequences at the last input position (10) when pretrained on the four tasks. The left plot shows the pretrained activations $t_{10}$ are not predictive of the completion. The right plot shows prefixes cluster the activations $t_{10}^{pt}$. Connecting the pretrained and prefixed activations highlights the bias. No dimensionality reduction is used; the clustering is solely due to the prefixes.

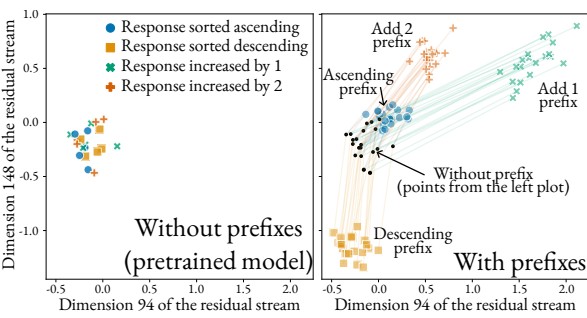

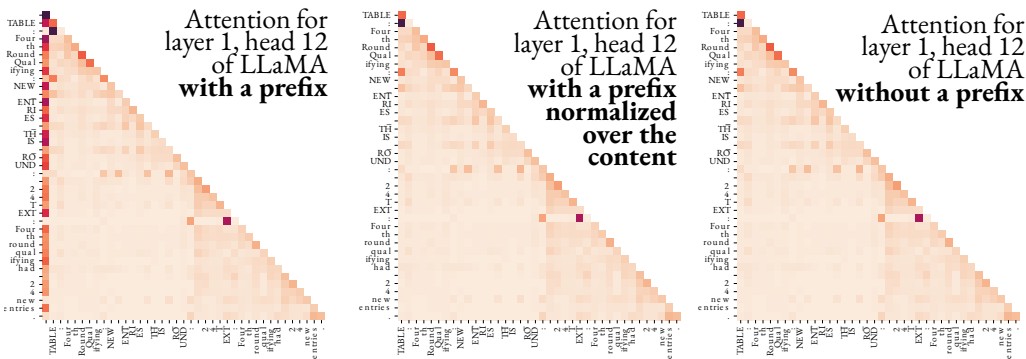

Figure 5: The attention of the twelfth head of the first layer of LLaMA (Touvron et al., 2023). The left plot shows the attention with a prefix of length one. The second plot shows the same attention but normalized such that the attenion over the non-prefix positions sums to 1. The right plot shows the attention of the pre-trained model (without prefix). The center and the right plots are the same, illustrating that the presence of the prefix indeed only scales down the attention over the content (non-prefix positions) but does not change its relative distribution, providing empirical validation of Equation (6). The test sequence is `TABLE: Fourth Round Qualifying :  NEW_ENTRIES_THIS_ROUND : 24 TEXT: Fourth round qualifying had 24 new entries.` from the DART table-to-test dataset (Nan et al., 2021).

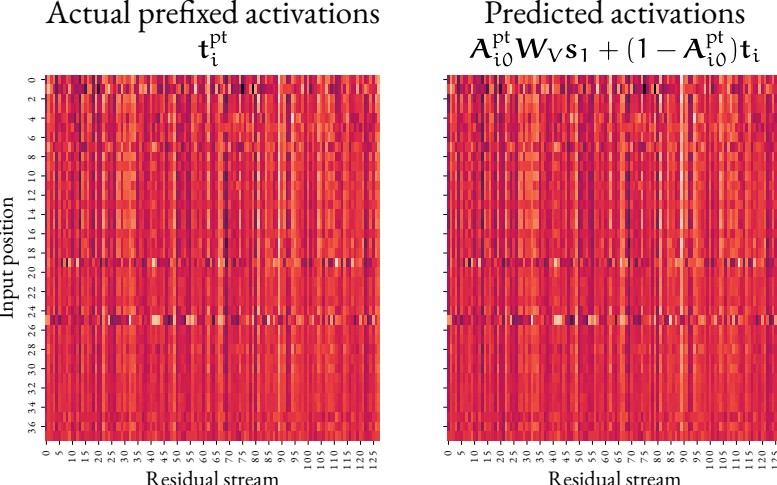

Figure 6: The activations of the twelfth head of the first layer of LLaMA (Touvron et al., 2023). The left plot shows the activations in the presence of the prefix. The right plot shows the activations $t_i$ of the pretrained model, scaled by one minus the attention that the prefix would take and then biased in the direction $W_V s_1$. The two plots are the same, illustrating that our theory, Equation (7) in particular, also holds for real-world large transformer models. The test sequence is the same as in Figure 5.

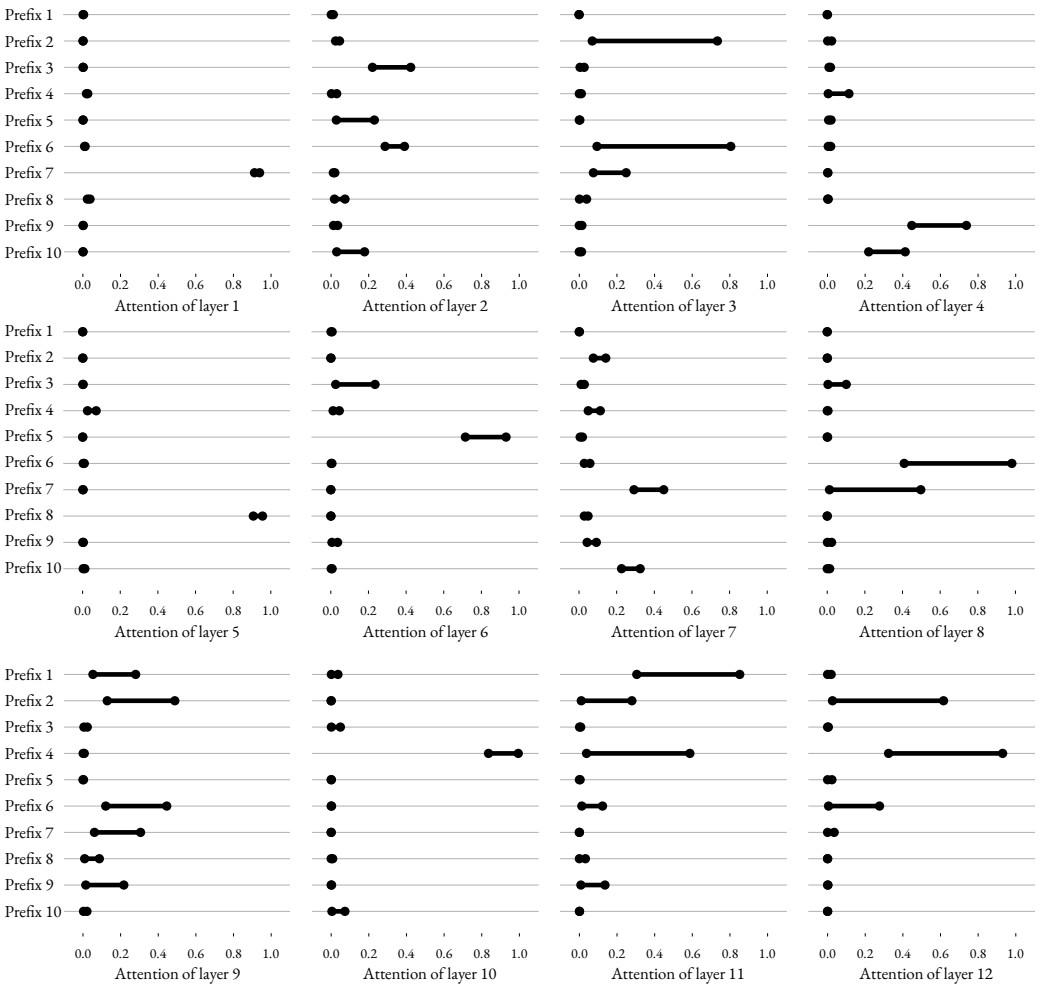

Figure 7: The range of attention (1st to 99th percentile) for a single GPT-2 (Radford et al., 2019) prefix trained on the Emotion dataset (Saravia et al., 2018). The prefix is of size 10 ($n_S = 10$). This is the attention of the last user input token ($n_X$) because this is the position at which the class prediction is done. For illustration purposes, we have normalized the attention so that the attention over the 10 prefix positions sums to 1. The range of attention over the 10 positions for each layer are shown.

## B  ATTENTION DISTRIBUTION OVER THE PREFIX

As discussed in Section 4, longer prefixes define a subspace from which the bias for the attention block is selected. For a prefix of size $n_S$, that means that this subspace is $n_S$-dimensional. Each of prefix position $j$ defines a basis vector $\boldsymbol{W}_V \boldsymbol{s}_j$ for this subspace, while the attention $\boldsymbol{A}^{\text{pt}}_{i,S_j}$ on this position determines how much of this basis component contributes to the bias.

In ordered to span the whole subspace and make full use of the capacity of the prefix, $\boldsymbol{A}^{\text{pt}}_{i,S_j}$ should vary between 0 and 1 for different inputs. However, we observe that this does not happen in practice. Figure 7 shows the ranges of attention the different prefix positions take for the GPT-2 model (Radford et al., 2019). For layer 1, for example, the attention each prefix positions gets is almost constant hence, the effective subspace is collapsed and there is a single bias vector that's applied to the attention layer output, regardless of the user input $X$.

Some other layers show slightly higher variation. For example, layer 3 has three prefix positions with large variations. Therefore, the effective bias subspace is 3-dimensional and the user input $X$ governs which bias vector from this subspace will be selected.

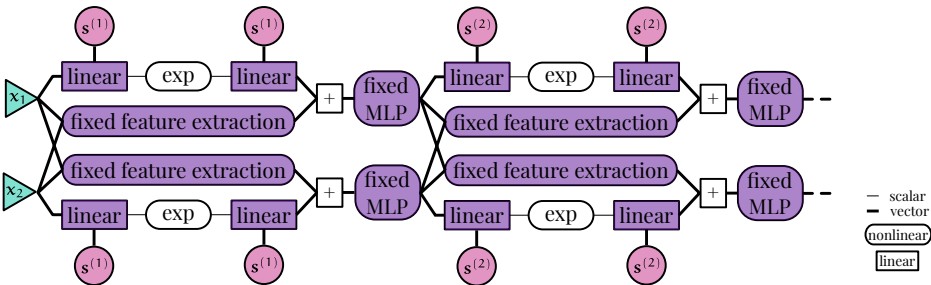

Figure 8: Prefix-tuning as a neural network architecture. While linearities and non-linearities are present, the only learnable parameters $s^{(1)}, s^{(2)}, ...$ have limited interaction with the inputs $x_1$ and $x_2$. The interaction of the prefix parameters with each input is only via the scalar attention, shown here with a light connection. The mixing of information between the inputs happens via residual connections with the pretrained fixed feature extraction and hence is not learnable. The MLP is also fixed and hence only acts as a multivariate activation function. This limited interaction explains why prefix-tuning struggles to learn new tasks even in deeper models.

## C EXPRESSIVITY OF PREFIX-TUNING ACROSS DEEPER MODELS

In Section 6, we considered the effect of the presence of the prefix in the first attention layer on the attention of the second. However, the effects become more complex as one adds more attention attention layers. We also ignored the MLPs between, but in practice they can play an important role. Here, instead, we analyse prefix-tuning as learning a neural network. We argue that while the resulting architecture includes both linear operation and non-linear activations, the structure is unlikely to learn efficiently.

For simplicity, we will consider two inputs $x_1$ and $x_2$ and a single prefix $s$. The output of the attention head, parameterized by $s$ is then:

$$
\begin{aligned}
\mathcal{A}_s(x_1, x_2) &= \langle y_1, y_2 \rangle \\
y_1 &= \frac{\exp(x_1{}^\top H s) W_V s + \exp(x_1{}^\top H x_1) W_V x_1 + \exp(x_1{}^\top H x_2) W_V x_2}{C_1} \\
y_2 &= \frac{\exp(x_2{}^\top H s) W_V s + \exp(x_2{}^\top H x_1) W_V x_1 + \exp(x_2{}^\top H x_2) W_V x_2,}{C_2}
\end{aligned}
\tag{11}
$$

where we have omitted the $^T/\sqrt{k}$ factors and have folded the softmax normalization into $C_1$ and $C_2$. The layer inputs, pretrained parameters and the learnable parameters are correspondingly highlighted. The attention head is clearly a non-linear operation. However, the learnable parameter $s$ participates only in the left term. It interacts with only one of the inputs at a time and only by computing a single scalar value $x_i^\top H s$. As we discussed above, the interaction between $x_1$ and $x_2$ is non-trainable and can be thought as a hard-coded feature extraction. Each of the outputs is then passed to a pretrained MLP which can be thought of as an activation function. This would be a *multivariate* activation function, which while unusual in the contemporary practice has been studied before (Solazzi and Uncini, 2004). Figure 8 illustrates the computation graph of the resulting neural network and shows that the only learnable interaction between the inputs is indirect. Therefore, prefix-tuning can be considered as learning a neural network where the only interaction between the inputs happens via non-learnable residual connections. Nevertheless, the alternating linear and nonlinear operations are reminiscent of the standard neural network architecture and their universal approximation properties (Hassoun, 1995). That begs the question if the prefix-tuning architecture can be a universal approximator and whether it would be a parameter-efficient one.

**An example of prefix-tuning failing to be a universal approximator.** While we leave the formal analysis of the representational capacity of prefix-tuning as future work, we provide an example of pretrained parameters for which the architecture is *not* a universal approximator. As can be seen in Figure 8, all information must pass through the non-learnable MLPs. Thus, if the MLPs destroy all input information, there is no value for the prefixes $s^{(1)}, s^{(2)}, ...$ that can change that fact. The MLPs

can destroy all information, if, e.g., one of their linear layers has a zero weight matrix. While this is unlikely to be learned with typical pretraining, this demonstrates that if prefix-tuning could be a universal approximator, that would pose specific requirements on the pretrained MLPs and it is not clear whether real-world pretraining would satisfy these requirements.

# D    EXTENDED RESULTS

In this appendix we present further experiments in the context of Section 5. We consider different prefix lengths ($n_S \in \{10, 50, 100\}$) and different model sizes (4, 16 and 32 layers). We consider two extensions, the first one maintains the pretraining step as in Section 5, the other one extends the set of pretraining tasks with 4 additional tasks. Both pretraining and prefix-tuning are done for 100 000 iterations with the prefix-tuning accuracy reported every 10 000 iterations.

## D.1    PRETRAINING AS IN SECTION 5

The first setting has the same pretraining tasks as in Section 5, namely sorting in ascending ($\nearrow$) or descending ($\searrow$) order, and adding one (+1) or two (+2) to each element of the input sequence. We evaluate by prefix-tuning on the same four tasks plus incrementing the ascending sorted sequence ($\nearrow$+1), double histogram ($\mathbb{H}$), element-wise modulo operation (with respect to the first element of the sequence), and FilterAtLeastNTimes which puts zeros at the positions of elements whose value appears at less than $N$ times in the sequence with $N$ being the first element of the sequence:

| | |
|---|---|
| Pretraining tasks: | Sort in ascending order ($\nearrow$)
Sort in descending order ($\searrow$)
Add 1(+1)
Add 2 (+2) |
| Prefix-tuning tasks: | Sort in ascending order ($\nearrow$)
Sort in descending order ($\searrow$)
Add 1(+1)
Add 2 (+2)
Sort ascending and add 1 ($\nearrow$+1)
Modulo the first element (Modulo)
Double Histogram ($\mathbb{H}$)
Filter the elements that are at least as large as the first element (FilterAtLeastNTimes) |

The results are plotted in Figure 9. As expected, the prefix-tuned accuracy on the pretraining tasks is close to 100%. Interestingly, prefix length 50 for the largest (32-layer) architecture appears to be an exception and does not learn the +1, $\nearrow$ and $\searrow$.

As also observed in Section 5, regardless of the prefix length and the model size, prefix-tuning a model pretrained with these four tasks cannot learn the double histogram task ($\mathbb{H}$). FilterAtLeast-NTimes also appears to be challenging but the 16-layer and 32-layer experiments reach about 50% accuracy for the longest prefix size $n_S = 100$. This is curious as FilterAtLeastNTimes is related to the double histogram task: it can be considered as thresholding the double histogram output. FilterAtLeastNTimes, similarly to double histogram, is therefore not compositional in the pretraining task. It is surprising then that FilterAtLeastNTimes would achieve higher accuracy than double histogram. This hints that, perhaps, compositionality does not fully explain why prefix-tuning works for some and not other downstream tasks.

The results in Figure 9 also hint that bigger is not always better. For example, the 4-layer model prefix-tuned for the Modulo task performs better with a prefix size 50 than the larger prefix size 100. A similar effect can be observed with the 16-layer model prefix-tuned for the $\nearrow$+1 task. Larger models are also not necessarily more conducive to successful prefix-tuning: for many of the cases the 32-layer models perform worse when prefix-tuned than the 16-layer models.

## D.2    EXTENDED PRETRAINING

The second setting extends the set of pretraining tasks. On top of the original three pretraining tasks $\nearrow$, $\searrow$,+1 (without +2) we also pretrain on element-wise modulo operation (with respect to the first element of the sequence), element-wise less than (less than the first element in the sequence),

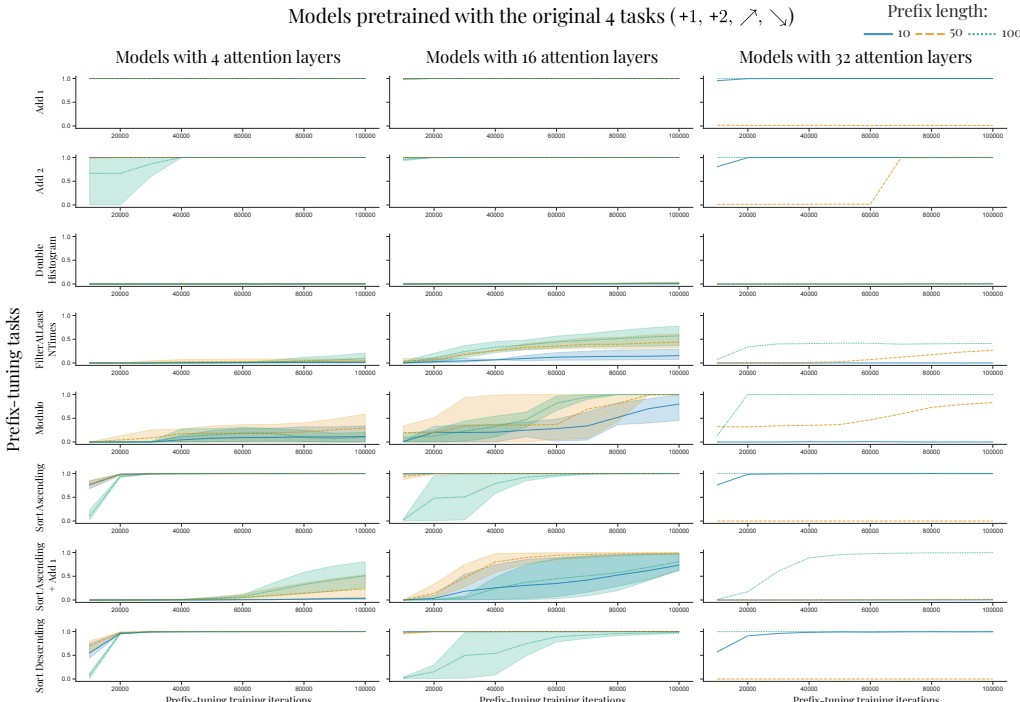

Figure 9: Extended result with pretraining as in Section 5. The pretraining and the prefix-tuning tasks are described in Appendix D.1. Each prefix was trained for 100 000 iterations and we report accuracy at every 10 000 iterations. Each experiment is performed with 3 random seeds, except the 32 layer case which, due to the computational costs involved, was performed only with one seed.

element-wise divisible (by the first element of the sequence), and inverse binary (element-wise negation). For the inverse binary task, the input is restricted to be binary. We evaluate by prefix-tuning on the same seven tasks, as well as 11 additional ones as listed in the following table:

| | |
|---|---|
| Pretraining tasks: | Sort in ascending order (↗)
Sort in descending order (↘)
Add 1 (+1)
Modulo the first element (Modulo)
Filter the elements that are less than the first element (LessThan)
Filter the elements that are divisible by first element (Divisible)
Element-wise negation (InverseBinary) |
| Prefix-tuning tasks: | Sort in ascending order (↗)
Sort in descending order (↘)
Add 1 (+1)
Add 2 (+2)
Add 3 (+3)
Modulo the first element (Modulo)
Filter the elements that are less than the first element (LessThan)
Filter the elements that are not less than the first element (MoreThanEqual)
Filter the elements that are divisible by first element (Divisible)
Filter the elements that are not divisible by first element (NotDivisible)
Element-wise negation (InverseBinary)
Double Histogram (ℍ)
Filter the elements that are at least as large as the first element (FilterAtLeastNTimes)
Sort ascending, followed by add 1 (↗+1)
Add 1, followed by LessThan (+1 + LessThan)
LessThan, followed by Add 1 (LessThan +1)
LessThan, followed by sort ascending (LessThan + ↗)
Divisible, followed by Add 1 (Divisible +1) |

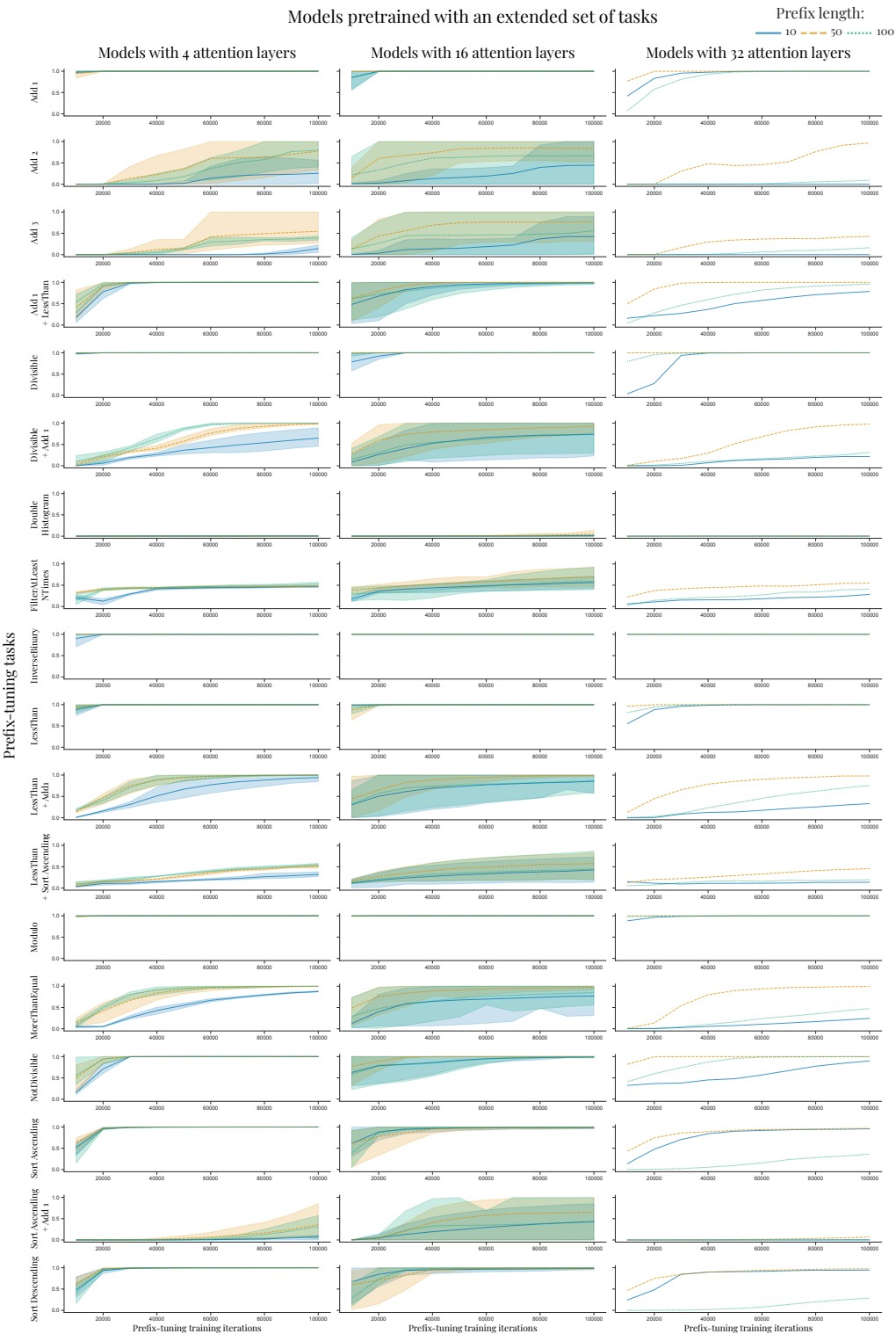

Figure 10: Extended result with pretraining with additional tasks. The pretraining and the prefix-tuning tasks are described in Appendix D.2. Each prefix was trained for 100 000 iterations and we report accuracy at every 10 000 iterations. Each experiment is performed with 3 random seeds, except the 32 layer case which, due to the computational costs involved, was performed only with one seed.

The results are shown in Figure 10. Even though the model did not see the +2 and +3 tasks, it appears that prefix-tuning can generalize from the +1 task. The +1 + LessThan task is another example of successful prefix-tuning for a task that is compositional in pretraining tasks. Similarly, for Divisible +1, LessThan +1, LessThan + ↗, MoreThanEqual, NotDivisible,

Similarly to the case in Appendix D.1, we observe several instances in which the largest, 32-layer, model performs worse, when prefix-tuned, than the smaller models, as well as cases where shorter prefixes perform better than longer ones.

# E    FURTHER EXPERIMENTS WITH MEMORIZATION

Our experiments in Section 5 focused on algorithmic tasks such as sorting, incrementing and counting. However, learning a natural language also includes a substantial memorization component. For example, learning a new language requires learning its vocabulary. Hence, if a novel task requires memorization of novel concepts, the fine-tuning method should be able to memorize them. To this end, we evaluate and compare the abilities of prefix-tuning and LoRA to learn to memorize a large number of new words and show that, for the same amount of learnable parameters, LoRA can learn to translate to a new language while prefix-tuning cannot. These findings further strengthen our results from Section 5.

We use the PHOR-in-One dataset (Costa et al., 2022) that contains 4921 unique translations of English words into German, Spanish and Portuguese. The dataset is preprocessed to remove all accents in order to ensure that fine-tuning does not require characters that the model has not seen during pre-training (e.g., *éçü* would become *ecu*). Character-level tokenization is used, with the additional `<TR>` token that separates the source and target words and a `<PAD>` token that we use to ensure all training sequences are of the same length.

We then pre-train a 4-layer 4-head transformer to translate the English words to German. As seen in Table 4, this model successfully memorizes 99.3% of the English-German pairs. We then want to fine-tune this model to instead translate the English words to Spanish. Spanish is linguistically closer to English than German, so, in a way, the fine-tuning task is simpler than the pre-training task.

We do prefix-tuning on English-Spanish pairs with a prefix of size $n_S = 66$ but it achieves only 0.18% accuracy (about 10 words which are the same in English and Spanish, e.g., *instrumental*, *orbital*, *solar*). However, rank-4 LoRA is able to achieve 94.8% accuracy with half the training iterations. Both fine-tuning methods have similar numbers of learnable parameters (see Table 4). Therefore, this is further evidence that prefix-tuning fails to learn a new task, while LoRA with the same number of learnable parameters can.

Note that there is no train-test split for the word pairs. We are evaluating the accuracy on the training set as this experiment is measuring *memorization* rather than *generalization*.

Table 4: Further experiments on memorization of word pairs form (Costa et al., 2022) in different languages. The model is pre-trained on the English-to-German word pairs to almost perfect accuracy. Prefix-tuning fails to modify it to memorize English-Spanish word pairs while LoRA with the same number of parameters and half the training iterations reaches 94% percent accuracy. 5 random seeds.

|  | Pre-training on English to German | Prefix-tune ($n_S$=66) on English to Spanish | Rank-4 LoRA on English to Spanish |
| --- | --- | --- | --- |
| Accuracy | 99.3 ± 0.1% | 0.18 ± 0.01% | 94.8 ± 1.1% |
| Learnable parameters | 3.19M | 67 584 | 66 780 |
| Training iterations | 50 000 | 100 000 | 50 000 |

