# OpenReview forum: "When Do Prompting and Prefix-Tuning Work? A Theory of Capabilities and Limitations"
_ICLR.cc/2024/Conference — ICLR 2024 poster_

### Official Review · Reviewer_rXHB · 2023-10-31

**Soundness:** 3 good
**Presentation:** 2 fair
**Contribution:** 4 excellent
**Rating:** 8
**Confidence:** 2

**Summary:**

The paper presents an analysis of how prompting, soft prompting, and pre-fix tuning work for transformer models and why these methods are not as capable as full-finetuning for transformer performance. The paper presents theorems on why soft prompting is able to elicit a wider range of behaviors than standard prompting, and prefix tuning than soft prompting. The paper then presents an explanation of why prefix-tuning cannot change the behavior of a transformer model as much as full fine-tuning. The paper then investigates why, if prefix-tuning is less powerful than fine-tuning, then why does it produce good results in practice? The paper posits that this result is due to prefix-tuning being very good for biasing a transformer toward performing pre-trained tasks and that the results in practice are a result of the model already understanding the task from pretraining.

**Strengths:**

The paper presents some very significant theoretical results and their implications for using transformer-based models. In particular, the idea that biasing the outputs through prefix (or even prompting or soft prompting) elicits pre-trained skills from the transformer model and that these pre-trained skills can be combined through this means, is important to really understanding both why techniques like prefix and prompting work, but also provide insight into the emergent behavior of models like LLMs. It also leads to the intriguing question of whether there is some basic set of tasks that something like an LLM needs to be pretrained on, in order to be able to practically accomplish just about any task in natural language.

The paper is also thorough in its investigation of the phenomenon of prefix-tuning and prompting by including both mathematical arguments for the claims made as well as simplified examples with actual code.

**Weaknesses:**

The paper has some clarity and soundness issues, from my reading. For clarity, I having trouble interpreting the attention figures in Figures 1, 3, etc. to see the patterns that the authors are trying to call out. Perhaps the image captions could include some interpretation guidance for the readers (e.g, the figures are meant to be read left-to-right, where …).

For soundness, there were a couple of areas, where I was not fully convinced of claims by the provided proofs. For Theorems 1 &2, I think can see why those are true, but having more of a sketch as to why they are true would improve both the clarity and firmly establish why soft prompting and prefix-tuning are more expressive in output generation than prompting. And in section 5.2, how does prefix-tuning change the attention of the next layer? Earlier on in the article, it is argued that changes to the attention layer have the form of $ W_{v} + \Delta W_{V}$ (i.e., equation 7), and yet the equations of section 5.2 do not have any alterations to $W_{V}$ or $H$. Rather it looks like the prefix-tuning changes the inputs to the next layer of attention rather than the attention block itself.

**Questions:**

In addition to the previously mentioned questions, in equation 7, what is the equivalence between the pre-trained ($t_i^{ft}$) and prefix-tuned ($t_i^{pt}$) model outputs, or is there one? While I generally by the argument that the change to $W_v$ means more changes to the outputs than adding the $W_V s_1$ term to the equation for the outputs, I am not convinced that that is always true or under what conditions it might achieve equivalence. Thus, to really cement the claim that pre-training holds more potential for outputs than prefix-tuning, it would be interesting to see where adding the bias term from prefix-tuning can be (and can’t be) equivalent to adding an update to $W_V$.

---

> ### Author Response · Authors · 2023-11-21
>
> Thank you for your insightful comments. Your recognition of the significance of our theoretical results and their implications for transformer-based models is greatly appreciated. We appreciate your acknowledgement of the thoroughness of our investigation into prefix-tuning and prompting and our practical code examples.
>
> > For clarity, I having trouble interpreting the attention figures in Figures 1, 3, etc. to see the patterns that the authors are trying to call out. Perhaps the image captions could include some interpretation guidance for the readers (e.g, the figures are meant to be read left-to-right, where …).
>
> Thank you for highlighting these clarity issues.
> We marked more clearly the part of the attention patterns that we refer to and elaborated further in the figure captions and the text in the revised version*. In Fig. 1 that would be the attention when the first response $Y_1$ is being generated, as this determines which task it going to be solved. For sorting in ascending order, when generating $Y_1$ we need to attend to the smallest values. And when sorting in descending order when generating $Y_1$ one should attend to the largest values. However, despite the prefix-tuning on the descending task, the model still attends to the smallest values as prefix-tuning cannot overcome the pre-trained attention pattern for sorting in ascending order. We also spotted a mistake in the labels in the "Model response" part of Figure 1 that we have now fixed. We believe that thanks to your feedback our paper is now much easier to read.
>
>
> > For Theorems 1 &2, I think can see why those are true, but having more of a sketch as to why they are true would improve both the clarity and firmly establish why soft prompting and prefix-tuning are more expressive in output generation than prompting.
>
> For both of these theorems, we provide proofs by construction. That is, we construct an example of a transformer model that attains the expressiveness described in the theorems. For Theorem 1, that is a model that can generate any target text unconditionally, and for Theorem 2, a model that can generate a target conditional token response to any token the user provides. In the Appendix, we extend this to a model that can generate any length conditional response to user input of any length. The Appendix provides the explicit constructions but also explanations as to how we came up with them and what the role of each individual component is.
>
> We also edited Section 3 and hope that it is much more clear now.
>
>
> > And in section 5.2, how does prefix-tuning change the attention of the next layer? Earlier on in the article, it is argued that changes to the attention layer have the form of $W_V + \Delta W_V$ (i.e., equation 7), and yet the equations of section 5.2 do not have any alterations to $W_V$ or $H$. Rather it looks like the prefix-tuning changes the inputs to the next layer of attention rather than the attention block itself.
>
> We see were the confusion may come from. One needs to distinguish between the changes with prefix-tuning $t^\text{pt}$ and the changes with full fine-tuning $t^\text{ft}$ in Eq. (7). An update of the form $W_V + \Delta W_V$ only happens in the full fine-tuning case. We cannot have $\Delta W_V$ in the prefix-tuning case as we cannot modify the model parameters. The equations in what is now Section 6.1 (Section 5.2 in the original version) refer to the prefix-tuned case (as the pt superscript shows). We also explain what happens in the full fine-tuning case in the following paragraph and show that $H$ is altered as $H + \Delta H$.
>
> Finally, you are absolutely right, the prefix-tuning changes the inputs to the next layer of attention rather than the attention block itself. This is exactly the message we were trying to convey: prefixing cannot change the parameters of the attention block but can affect its outputs by changing its inputs. We have modified the explanation and also added further discussion on the deeper effects in the new Section 6.2. We hope that has improved the clarity of this part of our work.
>
> --
>
> \* Note on the revised manuscript. We have implemented your feedback, added new experiments and improved the clarity of the writing. As a result the revised manuscript is currently 11 pages but we will reduce it to 9 for the camera ready version.

---

> ### Author Response · Authors · 2023-11-21
>
> > In addition to the previously mentioned questions, in equation 7, what is the equivalence between the pre-trained ($t^\text{ft}$) and prefix-tuned ($t^\text{pt}$) model outputs, or is there one? While I generally by the argument that the change to $W_V$ means more changes to the outputs than adding the $W_V s_1$ term to the equation for the outputs, I am not convinced that that is always true or under what conditions it might achieve equivalence. Thus, to really cement the claim that pre-training holds more potential for outputs than prefix-tuning, it would be interesting to see where adding the bias term from prefix-tuning can be (and can’t be) equivalent to adding an update to $W_V$.
>
> This is an excellent question and one that also interests us so much that we are currently working on a follow-up work!
>
> Let us give an example of why adding the bias term from prefix-tuning can never be equivalent to adding a general update to $W_V$.
> Consider that all inputs to the layer are on the surface of a sphere centered at the origin and that $W_V$ is a rotation matrix. Applying this rotation matrix would still keep all the points on the surface. Modifying $W_V$ to be $W_V'=W_V+\Delta W_V$, a different rotation matrix, will have the same property. However, there is no non-zero vector that one can add to a set of points on the surface of a sphere that will keep them all on the surface of the sphere. Therefore, this is a transformation that can be expressed with a matrix update (full fine-tuning) but not with a bias (prefix-tuning).
>
> An open question is whether there can exist particular pretrained transformers that can interpret this bias in a way that makes it much more expressive (though for a given transformer size it would still be less expressive than full fine-tuning). However, even if that would be possible, it would likely be very sample-inefficient and difficult to converge. Therefore, for all practical reasons, one would likely still be better using other fine-tuning approaches, e.g., LoRA. We added experiments that show that in Section 6.2 and Appendix C. We appreciate that this is a very interesting question so we extended the discussion on the deeper effects in the new Section 6. However, the full formal analysis is quite involved and of a very different flavor than the approach in this work, so we believe it is better suited as a separate follow-up work.

---

### Official Review · Reviewer_7Rcz · 2023-11-01

**Soundness:** 3 good
**Presentation:** 3 good
**Contribution:** 3 good
**Rating:** 6
**Confidence:** 3

**Summary:**

Context-based fine-tuning techniques like prompting, in-context learning, soft prompting, and prefix-tuning have gained popularity due to their ability to achieve good results with fewer parameters compared to full fine-tuning. However, we lack a theoretical understanding of how these methods affect the model's internal operations and their limitations. This paper reveals that while continuous embedding space is more flexible than discrete token space, soft prompting and prefix-tuning are less expressive than full fine-tuning. This means that techniques like prompting and in-context learning can leverage existing skills in a model but can't learn entirely new tasks that require different attention patterns. This understanding provides insights into the capabilities and limitations of these fine-tuning methods, helping researchers and practitioners make informed choices when applying them to various tasks.

**Strengths:**

S1. theoretical support to prompt learning is a pressing need and this paper provided a comprehensive view from theoretical analysis.

S2. I like their presentation, which is clear.

S3. their theoretical analysis is interesting.

**Weaknesses:**

Overall, this paper tried to solve a very interesting problem. I would be very happy to raise my score if the following concerns are addressed:


W1: prompting is not only used in linear data like text but also applied to non-linear data recently like graphs. It would be more solid to discuss them in the related work section (e.g. X Sun, et al. "All in One: Multi-task Prompting for Graph Neural Networks". KDD2023). It would be even better if the author could further confirm whether their theoretical analysis applies to the graph prompting area.

**Questions:**

see W1

---

> ### Author Response · Authors · 2023-11-21
>
> We appreciate your positive feedback on the clarity of our presentation and are glad to hear that you found our theoretical analysis interesting.
>
> > W1: prompting is not only used in linear data like text but also applied to non-linear data recently like graphs. It would be more solid to discuss them in the related work section (e.g. X Sun, et al. "All in One: Multi-task Prompting for Graph Neural Networks". KDD2023). It would be even better if the author could further confirm whether their theoretical analysis applies to the graph prompting area.
>
> Unfortunately, we are not very familiar with the graph prompting area.
> After reading the suggested paper, our impression is that it incorporates the prompts in an additive way (adding the prompt embeddings to the content embeddings or taking a dot product), rather than in the concatenation way typically used in language models. As a result, the graph prompt would affect the attention layers in a very different way.
> For example, we would not be able to separate the attention over the prompt positions and the attention over the content positions because the positions contain both prompts and content. For this reason, this approach falls out of the scope of our work.
>
> We tried to keep our analysis focused on the most general architecture to keep the results as broadly applicable as possible. That being said, as long as the attention mechanism works in a similar way, the limitations should also hold. For example, our results immediately translate to Visual Prompt Tuning (Jia et al., 2022).
>
> Regardless, given our limited expertise in graph transformers and graph prompting, it is difficult to provide more detailed answer. However, as graph problems admit a larger variety of attention architectures, we believe that this could be an interesting avenue for future work.
>
> Jia, M. et al. (2022). Visual Prompt Tuning. ECCV 2022.

---

> > ### Comment · Reviewer_7Rcz · 2023-11-22
> > **raise score**
> >
> > I thank the authors for their response. Although their response did not address my concern, their work is still very interesting and I raised my score from ''below'' to ''above''.

---

### Official Review · Reviewer_R11r · 2023-11-02

**Soundness:** 3 good
**Presentation:** 2 fair
**Contribution:** 3 good
**Rating:** 6
**Confidence:** 3

**Summary:**

This paper discusses the roles and limitations of context-based fine-tuning approaches (e.g. prefix fine-tuning) from a theoretical perspective. By analyzing the effects of attention mechanisms and computation within the model, the authors illustrate that there are structural limitations of prefix fine-tuning, while prefix fine-tuning is expressive due to continuous space. These limitations preclude prefix fine-tuning from learning new attention patterns, which makes it less expressive than full fine-tuning. The paper then reveals that the success of the context-based fine-tuning approach depends on the eliciting of skills in the pre-trained model, rather than learning new skills.

**Strengths:**

- This paper studies a valuable problem, which focuses on the capabilities and limitations of context-based fine-tuning, making a valuable contribution to the research community.
- The theoretical discussions effectively highlight the problems posed by context-based fine-tuning.

**Weaknesses:**

- It would be better to further test models larger than LLaMA-7B as models with more parameters may exhibit different properties.

- An additional ablation experiment may be required in subsection 'Prefix-tuning can combine knowledge from pretraining tasks to solve new tasks'. The authors utilize a 4-layer 4-head model to validate the point that prefix-tuning can learn a new task as long as the “skill” required to solve the new task is a combination of “skills” the pretrained model has seen. However, in a 4-layer 4-head model, a prefix-induced bias can have non-linear behavior when passed through non-linear MLPs and attention blocks as mentioned In Section 5.2. It would be better to further test a 1-layer 4-head model for further clarification.

- Minor grammar issues
There are also several minor grammar issues, just a few:
'However, generation generation is more interesting'
'This can be be clearly from the activations'

**Questions:**

-	Is the cross-layer effect enough to make the prefix fine-tuning learn new tasks when the transformer consists of more attention layers like many current language models?
-	Is there any potential improvements for tuning methods based on the analysis?

---

> ### Author Response · Authors · 2023-11-21
>
> We greatly appreciate your recognition of our work's focus on the capabilities and limitations of context-based fine-tuning, and are grateful for acknowledging its contribution to the research community.
>
> > It would be better to further test models larger than LLaMA-7B as models with more parameters may exhibit different properties.
>
> Our contributions are primarily theoretical and our experiments are simply a means to validating them. We use LLaMA-7B only to validate that a prefix cannot change the relative attention and only acts as a bias. We only look at this effect at the first layer as the non-linear effects kick in for the latter layers (we have elaborated further on this in Section 6 in the revised manuscript*). As our theory characterizes the effect of the prefix exactly, there would be no difference to this validation experiment if we look at larger models.
>
> > An additional ablation experiment may be required in subsection 'Prefix-tuning can combine knowledge from pretraining tasks to solve new tasks'. The authors utilize a 4-layer 4-head model to validate the point that prefix-tuning can learn a new task as long as the “skill” required to solve the new task is a combination of “skills” the pretrained model has seen. However, in a 4-layer 4-head model, a prefix-induced bias can have non-linear behavior when passed through non-linear MLPs and attention blocks as mentioned In Section 5.2. It would be better to further test a 1-layer 4-head model for further clarification.
>
> This is a great suggestion but we actually performed an even more illustrative experiment.
> The claim of that subsection is that prefix-tuning can combine pre-training skills but cannot learn completely new tasks.
> The "sort and increment" task is the one that can be solved via combining pre-training skills.
> We added a new task, "double histogram" (mapping each element to the number of elements in the sequence with the same value) which cannot be solved compositionally from the pre-training tasks.
> Prefix-tuning can reach very high accuracy on "sort and increment" but cannot learn "double histogram" at all.
> Therefore, this demonstrates that the ability of prefix-tuning to learn one but not the other task is not a matter of scale of the model but only due to the mismatch of the pre-training data and the fine-tuning task.
>
> > Is the cross-layer effect enough to make the prefix fine-tuning learn new tasks when the transformer consists of more attention layers like many current language models?
>
> We argue that, while it may potentially be possible for a very deep model with a lot of training to be able to learn some simple novel tasks, the sample complexity for this would be extremely high. Therefore, for all practical purposes, the depth does not help much. We think this is a really important question so we dedicated a new Section 6 in the revised manuscript on this question.
> We are also currently working on a followup work that formally examines the deeper effects and whether under some conditions prefix-tuning can have universal approximation properties.
>
> However, in Section 6.2 we added an experiment that shows that even if prefix-tuning has universal approximation properties, the sample complexity would be impracticably high, especially compared to other fine-tuning approaches. In particular, we show that prefix-tuning fails to learn the novel task "double histogram" but rank-1 LoRA with the exact same number of learnable parameters as the prefix easily reaches 92% accuracy. Therefore, even if prefix-tuning for deep models does have expressive capacity, it would be impractically difficult to optimize for and one would still be better using, e.g., LoRA. Fundamentally, learning is not only about density-type approximation results but also about the rate of convergence.
>
> > Is there any potential improvements for tuning methods based on the analysis?
>
> A potential improvement could be combining prefix-tuning with LoRA. Combining the two an studying whether prefix-tuning + LoRA is empirically better than LoRA. If that is the case, then this indicates that prefix-tuning could potentially have more value beyond just efficiency. This would be an interesting new direction to explore.
>
> --
>
> \* Note on the revised manuscript. We have implemented your feedback, added new experiments and improved the clarity of the writing. As a result the revised manuscript is currently 11 pages but we will reduce it to 9 for the camera ready version.

---

### Official Review · Reviewer_RrcA · 2023-11-06

**Soundness:** 3 good
**Presentation:** 3 good
**Contribution:** 3 good
**Rating:** 6
**Confidence:** 3

**Summary:**

The paper presents an insightful theoretical analysis of the limitations of context-based fine-tuning methods like prompting and prefix-tuning. While these methods have been empirically successful, the paper argues they are structurally less expressive than full fine-tuning and cannot learn tasks requiring new attention patterns. The paper tests these theoretical claims with minimal transformers and discusses the practical implications of these limitations.

**Strengths:**

+ The paper provides a valuable theoretical perspective on the limitations of prompting and prefix-tuning. It contributes to the understanding of how these methods compare to full fine-tuning in terms of their ability to learn new attention patterns, which is a significant contribution to the field.

+ The authors support their theoretical framework with empirical evidence. The use of minimal transformer models for testing provides clear illustrations of the theoretical limitations in a controlled experimental setting.

+ The paper's findings have practical relevance for the design of fine-tuning strategies in NLP applications. It helps practitioners understand when to employ prompting and prefix-tuning and when to opt for more expressive fine-tuning methods.

**Weaknesses:**

It would be beneficial to validate the theoretical findings with a broader set of experiments, including a variety of tasks, models, and datasets to confirm the universality of the proposed limitations.

A comparison with other fine-tuning methods such as transfer learning or domain-adaptive pretraining could provide a more comprehensive view of where prefix-tuning stands in the spectrum of fine-tuning techniques.

The paper identifies important limitations but does not provide detailed potential solutions or alternative methods that could overcome these limitations. Expanding on this could make the paper more impactful.

**Questions:**

The theoretical framework presented is compelling, but how does it hold up against the more recent transformer models that might use different mechanisms or have additional layers/heads?

How can the results inform the development of more efficient training procedures that could circumvent the limitations of prefix-tuning?

Could the authors provide a more detailed discussion on the practical implications of these findings for various NLP applications?

---

> ### Author Response · Authors · 2023-11-21
>
> Thank you for highlighting the theoretical insights and empirical evidence presented in our paper. We are also grateful for your recognition of the practical implications of our findings.
>
> > It would be beneficial to validate the theoretical findings with a broader set of experiments, including a variety of tasks, models, and datasets to confirm the universality of the proposed limitations.
>
> We completely agree and so we added a natural language task: learning to translate words between different languages. We pretrain on English to German translation with almost perfect accuracy. The novel task is translating English words to Spanish words. Prefix-tuning fails to learn this new task, while LoRA with the same number of parameters achieves 94% accuracy. While the experiments in the main part of the paper focused on algorithmic skills, this experiment demonstrates that a similar limitation holds for memorization tasks. Furthermore, the comparison with LoRA demonstrates that the limiting factor is not the number of learnable parameters, but where in the model architecture they are located. The full details are in Appendix C of the revised manuscript*.
>
> > A comparison with other fine-tuning methods such as transfer learning or domain-adaptive pretraining could provide a more comprehensive view of where prefix-tuning stands in the spectrum of fine-tuning techniques.
>
> This is an interesting suggestion! To the best of our understanding, while transfer learning and domain-adaptive pre-training are indeed separate pre-training techniques of themselves, the difference with classic pre-training is only in the data provided. Transfer learning and domain-adaptive pre-training refer to the particular choice of the training data but they still update all parameters. Their learning mechanism itself is full fine-tuning. Therefore, from the point of view of the expressiveness of the fine-tuning techniques, which is what this paper focuses on, the properties of transfer learning and domain-adaptive pre-training are exactly the same as the ones for prefix-tuning. We appreciate you highlighting this and will mention it in the paper.
>
> > The paper identifies important limitations but does not provide detailed potential solutions or alternative methods that could overcome these limitations. Expanding on this could make the paper more impactful.
>
> We do not propose new solutions because good solutions already exist. Our paper focused on identifying the limitations of prefix-tuning and prompting, which are context-based fine-tuning methods. Fine-tuning methods that do not have the expressiveness issues of prefix-tuning do exist but they require some sort of an update to the model parameters. For example, full fine-tuning or LoRA would readily address the expressiveness issues of prefix-tuning. Our paper aims to show that there is no free lunch and that the convenience of context-based fine-tuning methods comes with a cost in their expressiveness. Therefore, we argue that in certain cases one would be better off using the existing more expressive alternative methods. We realize we did not explicitly mention this in the manuscript so added additional experiments in Section 6.2 and Appendix C that show that LoRA with the same number of parameters can learn novel tasks that prefix-tuning struggles to.
>
> > The theoretical framework presented is compelling, but how does it hold up against the more recent transformer models that might use different mechanisms or have additional layers/heads?
>
> This would depend on the specific modification to the architecture. However, the number of heads does not affect our analysis in any way. Adding additional non-attention layers also does not have a major impact because prefix-tuning is applied only to attention layers. If you have a specific modification in mind, we could try to elaborate further on how our theoretical results would apply to it.
>
> > How can the results inform the development of more efficient training procedures that could circumvent the limitations of prefix-tuning?
>
> Our results essentially mean that as long as you are trying to learn a novel task, it is much more efficient to use, e.g., LoRA, instead of prefix-tuning. To this end, we have added an additional experiment in Section 6.2 in the revised version that illustrates this. It shows that prefix-tuning cannot learn the novel task "double histogram" but rank-1 LoRA with the exact same number of learnable parameters as the prefix easily reaches 92% accuracy. Therefore, circumventing the limitations of prefix-tuning can be trivially done by simply using other fine-tuning methods, such as LoRA.
>
> \* Note on the revised manuscript. We have implemented your feedback, added new experiments and improved the clarity of the writing. As a result the revised manuscript is currently 11 pages but we will reduce it to 9 for the camera ready version.

---

> ### Author Response · Authors · 2023-11-21
>
> > Could the authors provide a more detailed discussion on the practical implications of these findings for various NLP applications?
>
> There are several key implications of our findings:
>
> 1. If we know that certain tasks, versions of them, or components of them have been learned as part of the pre-training, prompting and prefix-tuning can be a very efficient way to specialize the pre-trained model for them.
>
> 2. If the target task is very different from the pre-training tasks, then prefix-tuning is not even remotely efficient fine-tuning method. In fact, as our experiments in Section 6.2 and Appendix C in the revised manuscript show, applying LoRA with the same number of parameters can be more effective in such cases. For example, the experiment that we added in Appendix C based on your suggestion shows that prefix-tuning cannot learn a new language but LoRA can. Prefix-tuning is also not likely to learn new subjects that it has not seen during pre-training if they require novel attention patterns, e.g., novel mathematical theories or new ways of formatting data.
>
> 3. In the Discussion section, we also mention positive implications for catastrophic forgetting and model alignment. While LoRA and full fine-tuning can result in catastrophic forgetting and erasing model alignment efforts, prefix-tuning and prompting are much less likely to exhibit such effects.

---

### Meta-Review · Area_Chair_MqTZ · 2023-12-12

**Metareview:**

This paper offers a theoretical analysis of the limitations of context-based fine-tuning methods, such as prompting, in-context learning, soft prompting, and prefix-tuning. It provides a theoretical understanding of how these methods impact the model's internal operations and their limitations. While these methods have been successful in practice, the paper argues that they are structurally less expressive than full fine-tuning and cannot effectively learn tasks requiring new attention patterns. This understanding sheds light on the capabilities and limitations of these fine-tuning methods, enabling researchers and practitioners to make informed decisions when applying them to various tasks. To further validate the theoretical findings, it would be beneficial to conduct a broader set of experiments, including a variety of tasks, models, and datasets.

**Justification For Why Not Higher Score:**

N/A

**Justification For Why Not Lower Score:**

This paper offers a theoretical analysis of the limitations of context-based fine-tuning methods, such as prompting, in-context learning, soft prompting, and prefix-tuning. It provides a theoretical understanding of how these methods impact the model's internal operations and their limitations. While these methods have been successful in practice, the paper argues that they are structurally less expressive than full fine-tuning and cannot effectively learn tasks requiring new attention patterns. This understanding sheds light on the capabilities and limitations of these fine-tuning methods, enabling researchers and practitioners to make informed decisions when applying them to various tasks.

---

### Decision · Program_Chairs · 2024-01-16

Accept (poster)